# Lawful kinematics link eye movements to the limits of high-speed perception

Martin Rolfs [1,2,3,6] ✉, Richard Schweitzer [1,3,6], Eric Castet [4], Tamara L. Watson[5] & Sven Ohl [1]

Perception requires active sampling of the environment. What part of the physical world can be perceived is limited by the sensory system's biophysical setup, but might be further constrained by the kinematic bounds of the motor actions used to acquire sensory information. Here, we tested this fundamental idea for humans' fastest and most frequent behavior—saccadic eye movements—which entail incidental sensory consequences (i.e., swift retinal motion) that rarely reach awareness in natural vision. Using high-speed video projection, we display rapidly moving stimuli that faithfully reproduce, or deviate from, saccades' lawful relation of velocity, duration, and amplitude. For each stimulus, observers perform perceptual tasks for which performance is contingent on consciously seeing the stimulus' motion trajectory. We uncover that visibility of the stimulus' movement is well predicted by the specific kinematics of saccades and their sensorimotor contingencies, reflecting even variability between individual observers. Computational modeling shows that spatio-temporal integration during early visual processing predicts this lawful relation in a tight range of biologically plausible parameters. These results suggest that the visual system takes into account motor kinematics when omitting an action's incidental sensory consequences, thereby preserving visual sensitivity to high-speed object motion.

Research in humans and across the animal kingdom shows that sensory input is contingent on active behavior. Sampling actions like whisking and sniffing, touching and looking control the flow of information to the brain[1–7]. Sensory processes might thus best be understood in the context of ongoing movements of the corresponding sensory organ[8–10]. Indeed, when sampling actions are both frequent and give rise to specific, reliable sensory consequences, the perceptual system might be wrought specifically to deal with information that these actions impose on the sensory surface[9,10]. In the extreme, the very limits of a sensory system's access to the physical world might be defined not just by biophysical constraints, but further curtailed by the kinematic bounds of the motor actions that acquire sensory information. Conclusive demonstrations of such action-dependence of the limits of perception are missing, but a key prediction is that perceptual processes should be tuned to an action's typical sensory correlates, even in the absence of the accompanying action[11–13]. Here we confirm this prediction for a fundamental perceptual process in human vision: We demonstrate that a shared law links the limits of perceiving stimuli moving at high speed to the sensory consequences of rapid eye movements.

Human vision is a prime example of the tight coupling between action and perception. Rapid eye movements called saccades shift the high-resolution fovea to new locations in the visual scene, affording access to fine visual detail at that location during the next gaze fixation.

[1]Department of Psychology, Humboldt-Universität zu Berlin, Berlin, Germany. [2]Bernstein Center for Computational Neuroscience Berlin, Berlin, Germany. [3]Exzellenzcluster Science of Intelligence, Technische Universität Berlin, Berlin, Germany. [4]Centre de Recherche en Psychologie et Neurosciences, Aix-Marseille Université, CNRS, Marseille, France. [5]School of Social Sciences, Western Sydney University, NSW Sydney, Australia. [6]These authors contributed equally: Martin Rolfs, Richard Schweitzer. ✉e-mail: martin.rolfs@hu-berlin.de

Saccades provide an ideal test case for the action-dependence of perception: They are the most frequent movement of the human body, occurring some 10,000 times every waking hour, and they have reliable, stereotyped kinematics[14] that impose systematic sensory consequences on the retinal surface[10,15]. Most prominently, the *main sequence* describes a lawful relation of saccadic speed and duration to the movement's amplitude: both peak velocity and duration of the movement increase systematically with the distance the eyes travel[16] (Fig. 1a). This relation is lawful in that the relevant kinematic variables (amplitude, peak velocity, and duration) mathematically relate to each other[17–19], and it applies across all known species that make saccadic movements (even including fruit flies[20]). Critically, every movement of the eyes with respect to the world yields an instantaneous, equal and opposite movement of the world projected on the retina. As a consequence, saccades entail rapid shifts of the retinal image that obey the same main-sequence relation: With increasing amplitude, saccades will result in larger movements and higher speeds of the retinal image (Fig. 1b). Even though visual processing remains operational during saccades[21–31], this saccade-induced retinal motion is subjectively invisible during natural vision—a phenomenon referred to as saccadic omission[32–34]. A broad range of accounts has been put forward for this reliable absence of perceiving the sensory consequences of saccades[10,35,36], invoking mechanical[37,38], retinal[22,22,39,40], and extraretinal mechanisms[29,41,42]. While there is consistent evidence for the saccade-locked reduction of visual sensitivity (especially to low spatial frequencies[42–44] which should remain visible at high saccadic speeds[45]) perception of motion during saccades is well possible within a resolvable temporal-frequency range[21,23,31]. These findings can be reconciled by a constant visibility threshold at some temporal frequency[46,47] (or speed, for a given spatial frequency[22,45,48]) beyond which a stimulus becomes invisible. Saccades often have small movement amplitudes that induce lower retinal velocities, thereby allowing for a broader range of resolvable spatial frequencies[15], yet the visual consequences of these eye movements are routinely omitted from conscious perception. We thus investigated if the limits of visibility of stimuli at high speed are predicted by saccade-related metrics, specifically those described by the main sequence. Such a relationship would suggest that the kinematics of the retinal image, lawfully induced by saccades, could shape the profile by which moving stimuli are omitted from perception.

The kinematic properties of eye movements and their perceptual correlates present a unique opportunity to investigate whether perceptual processes relate to the regularities that actions impose on the sensory input. As saccade kinematics routinely produce extremely fast pulses of motion on the retina, we predicted that their lawful sensory consequences relate to the perceptual limits for stimuli moving over finite distances at high speed. To test this idea, we used high-speed video projection to reproduce the lawful conjunction of saccade speed, duration, and amplitude in a moving visual stimulus presented during gaze fixation. We show across five experiments (with pre-registered analyses and predictions; see "Methods") that its visibility is well-predicted by the sensory consequences of saccades, reflecting even inter-individual variations in eye movement kinematics. As a consequence, perception reflects a fine compromise between sensitivity to high-speed stimuli[45] and omission of finite motion consistent with saccades. We show that this compromise is captured by a parsimonious model of early visual processes. Our results make a strong case that the functional and implementational properties of visual processing are fundamentally aligned with the consequences of oculomotor behavior.

## Results

We developed a simple psychophysical paradigm to assess observers' ability to see a stimulus at high speeds (Fig. 2a). A high-contrast vertical grating (Gabor patch) appeared on one side of the screen (left or right of fixation), rapidly moved to the other side, and then disappeared again. We varied the stimulus' movement amplitude $A$ between 4 and 12 dva (Fig. 2b). Its horizontal movement speed (*Absolute speed v*; Fig. 2c) was based on the expected peak velocity $v_p$ of a horizontal saccade for any given amplitude[18,19], multiplied by a factor of 0.25 to 1.25 for each amplitude (*Relative speed $v_{rel}$*; Fig. 2d). Movement duration resulted from the combination of movement amplitude and absolute speed ($D = A/v$) and occupied the central portion of the 500 ms stimulus duration (the stimulus was stationary before and after the movement; Fig. 2d). While the stimulus moved rapidly, tight fixation control at the center of the screen ensured that observers did not execute saccadic eye movements throughout stimulus presentation.

To enable measurement of stimulus visibility, we used two different tasks. In Experiment 1, the motion path curved (akin to saccades[49]) slightly upwards or downwards, following a circular segment (Fig. 2b), and observers judged the vertical component of the moving stimulus in a direction-discrimination task (up vs down). In Experiments 2a and 2b (presented in detail in Supplementary Note 2 and Supplementary Note 3), the high-speed stimulus followed a straight horizontal path, and observers had to distinguish movement-present from movement-absent trials in a detection task. All critical aspects of the results replicated across tasks, and we used the direction-discrimination task (Exp. 1) as the basis for all subsequent experiments.

Depending on its speed, the moving stimulus gave rise to two qualitatively distinct percepts: At slower speeds, observers perceived the grating as moving smoothly from one side to the other. At higher speeds, the movement—though physically continuous—was no longer visible to the observer, who instead perceived the stimulus as jumping from its initial to its final position, such that continuous motion was phenomenologically indistinguishable from a simple displacement (apparent motion percept; see also Supplementary Note 2 and Supplementary Note 3). The transition from a continuous to an apparent motion percept renders the movement path indiscriminable. We refer to the speed at which this transition occurs as the visibility threshold.

This paradigm allows for clear, mutually exclusive predictions: If visibility thresholds were simply a function of absolute movement

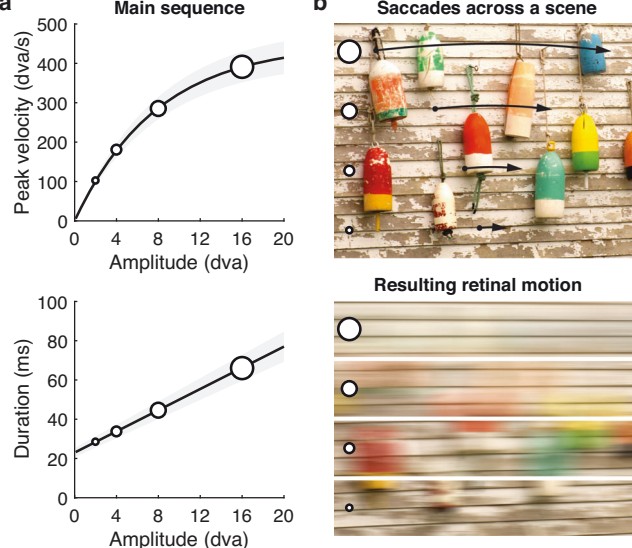

**Fig. 1 | Schematic of the main-sequence relation of saccadic eye movements and their instantaneous sensory consequences. a** Peak velocity and duration of saccades increase lawfully with movement amplitude. **b** Saccades impose motion on the retina with kinematics that follow the same conjunction of amplitude, speed, and duration (illustrated for saccade amplitudes of 2, 4, 8, and 16 degrees of visual angle, dva).

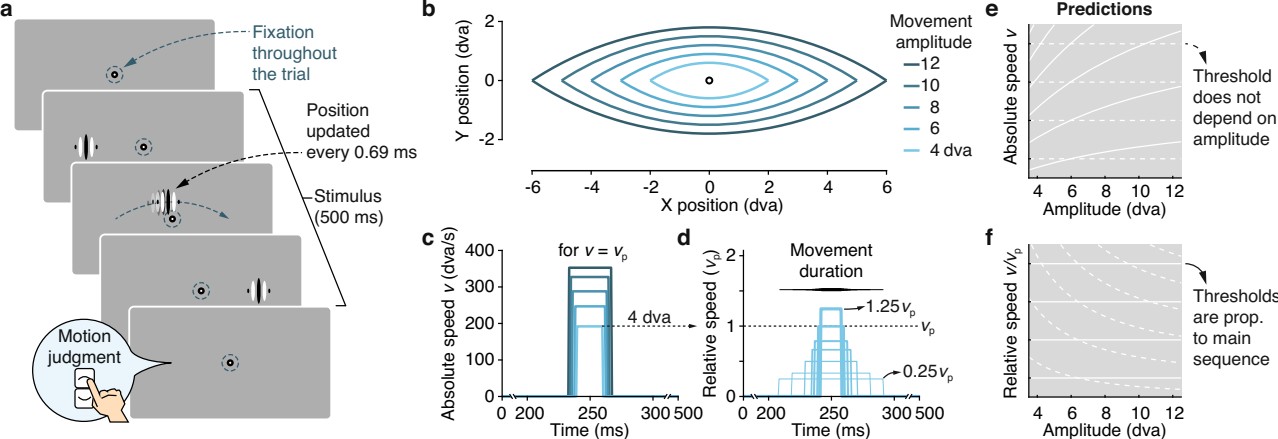

**Fig. 2 | Assessing visibility of stimuli moving over finite amplitudes at high speeds. a** Participants fixated throughout a trial as a vertical Gabor stimulus (100% contrast, 1 cycle/dva, 1/3 dva envelope) appeared on one side of the screen, rapidly moved to the opposite site of fixation, and then disappeared again. The stimulus' path followed a circular arc (height: 15% of the horizontal amplitude) and observers judged the vertical direction in a 2-alternative forced choice (upward vs downward). While the stimulus moved rapidly, strict fixation control at the center of the screen ensured that observers maintained fixation throughout the 500 ms of stimulus presentation. We updated stimulus position at 1440 Hz (i.e., every 0.69 ms) to achieve smooth, continuous motion and reliable timing at all speeds (see Supplementary Methods: Faithful rendering of high-speed motion). **b** Trajectories of stimulus motion for the five amplitudes tested. **c** Absolute speed $v$ of the stimulus as a function of time, shown for the peak velocity $v_p$ corresponding to each movement amplitude. Movement duration varied as a function of movement amplitude and speed ($D = A/v$); the stimulus was stationary at its start and end points before and after the movement. **d** For each movement amplitude (here, 4 dva), we varied the absolute speed in proportion to $v_p$ (each line corresponds to one relative movement speed, $v_{rel}$). Prediction of thresholds expressed for absolute (**e**) and relative (**f**) speed when these thresholds are invariant to movement amplitude (dashed white lines) or, alternatively, proportional to the main sequence (solid white lines).

speed ($v$), then they should be independent of movement amplitude (Fig. 2e, f; dashed white lines). The rationale for this prediction is that, because of the fixed spatial frequency used in our experiments, speed is equivalent to temporal frequency, which predicts visibility thresholds for a wide range of visual stimuli[21,45–48] (see "Introduction"). In contrast, if thresholds are a function of the kinematics of retinal motion during saccades, then it should systematically increase with movement amplitude, in proportion to the main-sequence. In that case, relative movement speed, expressed with respect to the expected peak velocity of a saccade ($v_{rel} = v/v_p$), should determine visibility (Fig. 2e, f; solid white lines).

### Visibility emulates the saccadic amplitude-velocity relation

Observers' performance (i.e., their ability to report the vertical component of the stimulus trajectory) transitioned from close to perfect at the slowest speeds to chance level at the highest speeds (Fig. 3a). We captured this relation by fitting negative-slope psychometric functions. Using hierarchical Bayesian modeling, we determined visibility thresholds for all movement amplitudes and individual observers in a single model[50], and obtained 95% credible intervals (CIs) to evaluate the impact of movement amplitude on visibility thresholds.

We first analyzed performance as a function of absolute movement speed ($v$, expressed in dva/s; Fig. 3a, left). Performance systematically increased with movement amplitude, shifting the psychometric function to the right, such that higher absolute speeds were visible when the stimulus moved over larger distances. Accordingly, visibility thresholds increased monotonically as a function of movement amplitude (Fig. 3a, right). This pattern of results was remarkably consistent across observers (gray lines) and closely followed the prediction based on the main sequence (solid white lines), violating the prediction based on an absolute speed threshold (dashed white lines). To assess if thresholds indeed show non-linear trends as a function of movement amplitude (consistent with the main sequence), we reparameterized threshold parameters as orthogonal polynomial contrasts (see "Methods" for details). In addition to a clear intercept, for which CIs did not include zero, we found evidence for linear and

terms, as well as a small cubic trend (Table S1, Absolute speed, in Supplementary Note 1). While the parameter estimates for the linear and cubic terms were positive, the coefficient of the quadratic term was negative. These results capture the decelerating increase of thresholds as a function of movement amplitude. The highest degree (quartic) term was not significant.

We next analyzed performance as a function of relative movement speed ($v_{rel}$, expressed in units of $v_p$). When expressed this way (Fig. 3b, left), psychometric functions collapsed entirely, such that thresholds settled around 53% of saccadic peak velocity irrespective of movement amplitude (Fig. 3b, right). This striking result corroborates the prediction that thresholds are proportional to the main-sequence relation of saccades (solid white lines). Orthogonal contrasts confirmed this amplitude-independence as only the CI of the intercept did not include zero, whereas all polynomial trends did (Table S1, Relative speed, in Supplementary Note 1). Visibility thresholds were thus better predicted by the main-sequence relation of saccades than by any polynomial variation.

### Replication in detection tasks and for time-varying velocity

To control for a potential impact of the curvature discrimination task on our results (e.g., due to the vertical motion component), and to understand the role of the velocity profile of the stimulus, we replicated these pivotal results in two supplementary experiments (see Supplementary Note 2 and Supplementary Note 3 for detailed results). Specifically, we developed a detection task (Fig. 4a), in which we eliminated the vertical component of the stimulus' movement trajectory. That is, instead of discriminating the curvature of the stimulus' motion path, observers now distinguished the presence or absence of continuous motion on a straight horizontal path.

In Experiment 2a, we moved stimuli at constant speed, as in Experiment 1 (Fig. 2c, d). By fitting Naka-Rushton functions to observers' sensitivity ($d'$) as a function of stimulus speed, we extracted visibility thresholds as the speed at half the asymptotic performance. While performance in this task was lower than in the curvature discrimination task (Exp. 1), visibility thresholds (Fig. 4b) closely followed

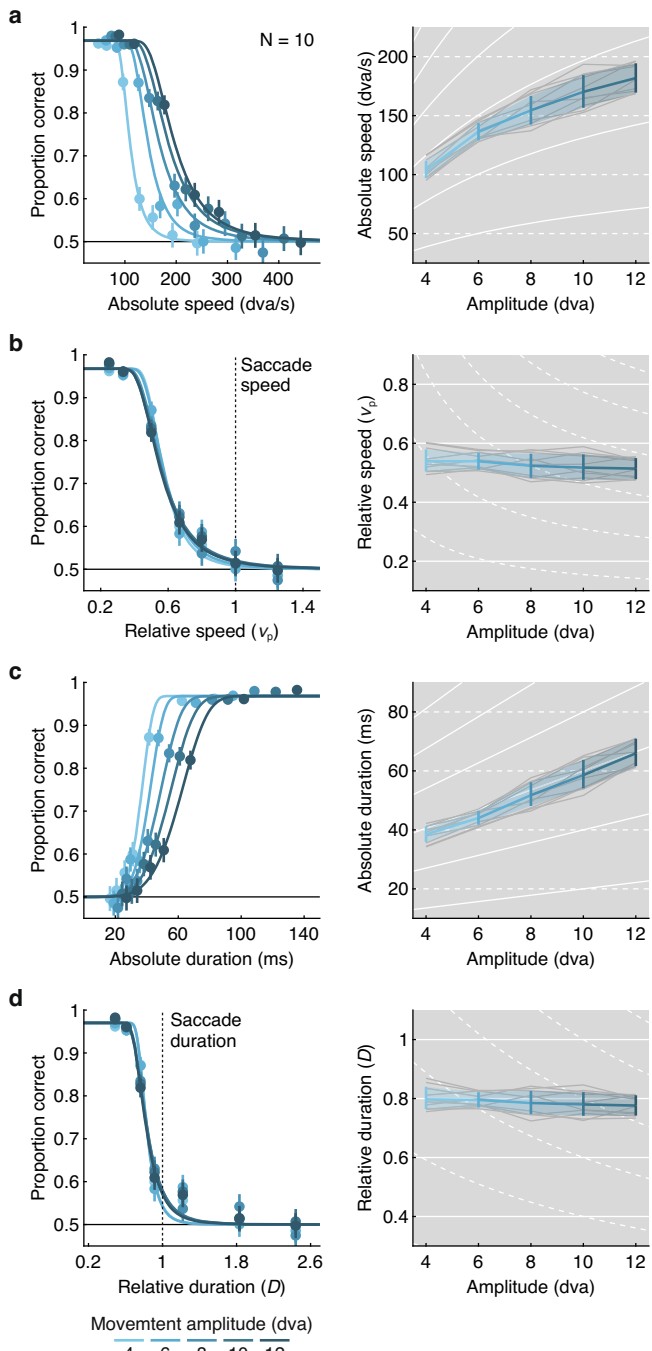

**Fig. 3 | Visibility depends on the conjunction of movement amplitude, speed, and duration, emulating the main-sequence relation of saccadic eye movements.** Performance (left column) and visibility thresholds resulting from fitted psychometric functions (right column) as a function of **a** absolute movement speed, **b** relative movement speed (1 corresponds to the expected speed of a saccade of a given amplitude), **c** absolute movement duration, and **d** relative movement duration (1 corresponds to the expected duration of a saccade of a given amplitude), as well as its amplitude. Data were obtained from $N = 10$ observers completing a total of 39,745 trials. The modes of estimated threshold parameters of the psychometric functions are shown as a function of movement amplitude both averaged across observers (blue), and for individual observers (gray). Error bars (left) are 95% confidence intervals across observers; error bands (right) are 95% credible intervals across observers. Solid white lines in the background indicate predictions in which thresholds depend on movement amplitude, proportional to the main sequence; dashed white lines indicate predictions in which thresholds are independent of movement amplitude.

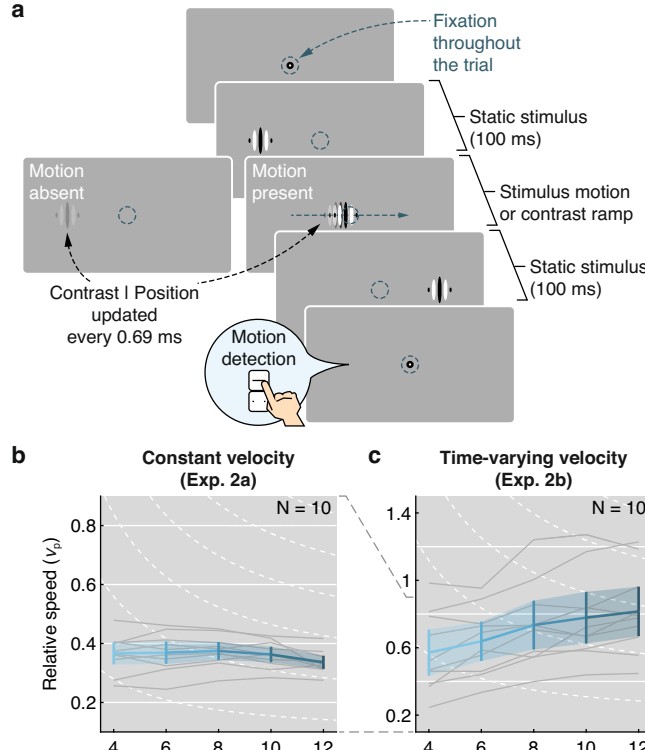

**Fig. 4 | Visibility of high-speed motion in a detection task. a** Trial procedure. Observers judged whether the stimulus jumped (Motion absent) or moved continuously on a straight horizontal path (Motion present) from its initial to its final location. **b, c** Visibility thresholds in Experiment 2a (using constant velocity) and Experiment 2b (using time-varying velocity profiles) as a function of relative movement speed, plotted for each amplitude. Data were obtained from $N = 10$ observers each completing a total of 68,989 trials and 40,937 trials in Experiments 2a and 2b, respectively. Note the different scale highlighted by dashed gray lines between (**b**) and (**c**). Conventions as in Fig. 3a, b.

the prediction based on the main sequence (solid white lines), with remarkable consistency across observers (gray lines). These results confirm that the vertical motion component was not a critical factor for the observed pattern of results in Experiment 1.

In Experiment 2b, we moved stimuli with time-varying speed (Fig. 4c), including a rapid acceleration phase and a slower deceleration phase, modeled after the velocity profile of saccades[18]. Critically, the slow deceleration has longer duration for larger saccade amplitudes and renders saccadic omission imperfect[34]. Consistent with this finding, we predicted that (in addition to the dependence of visibility on the main-sequence relation) detection of motion will increase slightly with movement amplitude. Indeed, while on average performance was higher compared to a constant velocity profile, visibility thresholds showed this predicted dependence on movement amplitude and speed (Fig. 4c). Again, this result was remarkably consistent across observers (gray lines).

Together, Experiments 2a and 2b confirmed that visibility thresholds systematically depend on movement amplitude in a way well-described by the main-sequence relation of saccadic eye movements, and that this result is not dependent on the curvature task. They also underline that our results are relevant to saccadic omission[32,34]: While saccades impose time-varying velocity profiles onto the retinal surface, key kinematic parameters of saccades—amplitude, velocity, and duration—suffice to predict visibility of stimulus motion.

## Visibility emulates the saccadic amplitude-duration relation

A critical parameter for detection of motion is its duration, and the integration time required for detection decreases with stimulus speed[51–54]. For any given speed, therefore, larger movement amplitudes may provide a performance advantage in our task, as durations scale linearly with movement amplitude ($D = A/v$). However, if visibility thresholds emulate the amplitude-duration relation of saccades, we can make a strong prediction: Smaller movement amplitudes should require shorter movement durations to be visible than larger movement amplitudes would (Fig. 1a, bottom). To investigate the impact of movement duration on stimulus visibility, we reanalyzed performance in Experiment 1 as a function of movement duration, using the same approach as for stimulus speed.

Performance increased as a function of absolute movement duration ($D$, expressed in dva/s; Fig. 3c, left). Yet this was not merely a consequence of increasing time for motion integration. Performance systematically increased with decreasing movement amplitude, shifting the psychometric function to the left. Indeed, the shorter the motion path, the shorter movement durations could be to render the stimulus visible. Accordingly, visibility thresholds increased monotonically as a function of movement amplitude (Fig. 3c, right), closely following the linear prediction based on the main sequence (solid white lines), and violating the prediction based on an absolute duration threshold (dashed white lines). The linear term of thresholds was clearly positive whereas higher-order terms were not significant (Table S1, Absolute duration, in Supplementary Note 1).

Expressing performance as a function of relative movement duration (i.e., normalizing absolute duration by the expected duration of a saccade, $D_{rel} = v_p D/A$), psychometric functions again collapsed (Fig. 3d, left). Thresholds were around 80% of saccade durations (Fig. 3d, right) and no longer depended on movement amplitude (Table S1, Relative duration, in Supplementary Note 1), confirming that they are proportional to the amplitude-duration relation of saccades.

Together with the analyses as a function of movement speed, these results paint the consistent picture that the visibility of stimuli moving over finite distances is best predicted by a lawful conjunction of movement speed, amplitude and duration. Intriguingly, this conjunction is exactly proportional to the main sequence that describes the kinematics of saccadic eye movements.

## Visibility covaries with saccade kinematics

While the functional form of the main sequence is consistent across the population[17,55,56], its parameters can vary within individuals (e.g., across movement directions) and, more considerably, between individuals[55,57–60] with high reliability across experimental conditions[56,59]. We capitalized on these reliable sources of variability to investigate possible links between oculomotor kinematics and visibility of high-speed stimuli, extending our protocol from horizontal (left, right) to vertical movement directions (up, down), first for a range of movement amplitudes (4–12 dva) in a small sample (Experiment 3, $N = 6$), and then for a single amplitude (8 dva) in a larger sample (Experiment 4, $N = 36$). For each movement direction, performance was a conjunctive function of movement amplitude, speed, and duration defined by the standard main sequence (Exp. 3), generalizing the findings of Experiment 1 across the cardinal movement directions. Importantly, in both experiments, visibility thresholds also varied systematically across movement directions and, more considerably, across individuals (see Supplementary Note 4, Fig. S4a, b).

To relate this variability to individual observers' eye movement kinematics, we also recorded visually-guided saccades (4–12 dva; left, right, up, down) in separate blocks of trials, and used a biophysical model of eye movement kinematics to isolate the relevant eyeball velocity from the recorded pupil velocity[61,62]. This provided an estimate of each individual's amplitude, speed, and duration of retinal

motion during saccades (see "Methods", Analysis of saccade kinematics; Supplementary Note 4, Fig. S4c, d). We reasoned that if visibility of high-speed stimuli is indeed related to the visual system's constant exposure to saccade-imposed motion, we may find covariations between individual visibility thresholds (during fixation) and corresponding eye movement kinematics. Critically, the retinal motion caused by saccades is equal in amplitude but opposite in direction to the eye movement itself. Our (pre-registered) prediction was, therefore, that the speed and duration of saccades of the same amplitude but opposite direction of motion (henceforth, the saccade's retinal direction) should best predict the corresponding visibility thresholds. For example, the visibility thresholds for downward motion should be predicted better by the speed and duration of upward saccades (with downward retinal direction) than downward saccades (with upward retinal direction), and vice versa.

Despite the fact that peak velocity was highly correlated between saccades in opposing directions (Experiment 3: $\rho = 0.816$, $p < 0.001$; Exp 4: $\rho = 0.425$, $p < 0.001$), perceptual speed thresholds in both experiments showed higher correlations (side panels in Fig. 5a) with the peak velocity of saccades whose retinal (Exp. 3: $\rho = 0.833$, $p < 0.001$; Exp. 4: $\rho = 0.366$, $p = 0.001$; Fig. 5a) rather than spatial direction (Exp. 3: $\rho = 0.682$, $p < 0.001$; Exp. 4: $\rho = 0.125$, $p = 0.137$) matched the stimulus' motion direction. Note that significant

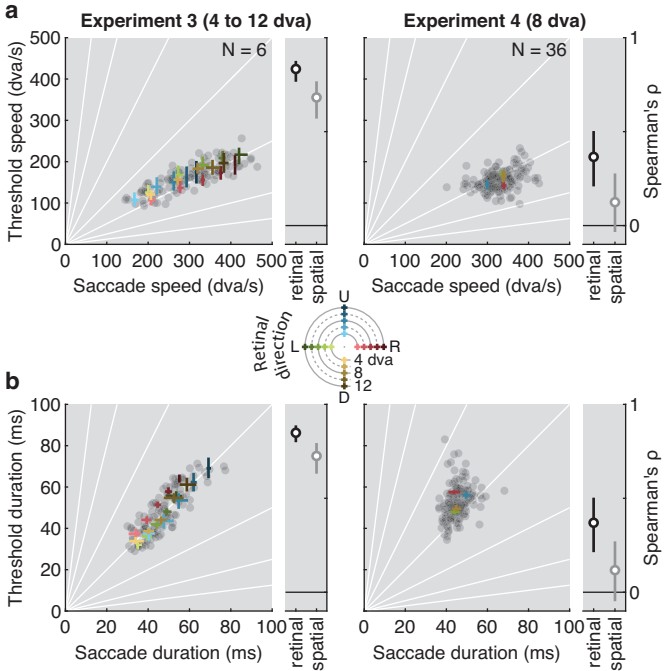

**Fig. 5 | Visibility thresholds and saccade kinematics covary across amplitudes, directions, and individuals.** Correlation between visibility thresholds (obtained from individuals' psychometric functions for a given amplitude and direction) and saccade kinematics (obtained from the prediction of individuals' main sequence for a given amplitude and direction) for **a** absolute movement speed and **b** absolute movement duration in Experiment 3 (left) and Experiment 4 (right). Data were obtained from $N = 6$ and $N = 36$ observers completing a total of 48,553 trials and 47,817 trials in Experiments 3 and 4, respectively. Each gray dot is one combination of observer, direction (Experiments 3 and 4), and amplitude (Experiment 3). Colored data points show the mean across observers for each combination of amplitude and direction. To facilitate comparison, thresholds are plotted against the corresponding kinematic of the saccade in the opposite direction, matching retinal motion direction. Rank correlations (Spearman's $\rho$) are shown in side panels. Matching spatial (gray) rather than retinal direction (black) provides a baseline for comparison. Error bars are bootstrapped 95% confidence intervals across observers. Solid white lines in the background indicate correlations with zero-intercept and various slopes.

correlations in Experiment 3 were not solely driven by the main sequence itself: Even after removing the main effect of saccade amplitude by removing the across-subjects mean of each of the dependent variables, we still found that saccadic peak velocity correlated significantly with visibility thresholds for absolute movement speed in the opposite (retinal) direction ($\rho = 0.40$, $p < 0.001$), whereas the correlations for the spatial direction vanished ($\rho = -0.07$, $p = 0.466$). Linear regressions confirmed that the prediction of speed thresholds from saccade peak velocity was superior for saccades matched with respect to their retinal rather than their spatial direction both in Experiment 3 (retinal: $R^2 = 0.68$, $\beta = 0.363$, SE = 0.023, $t = 15.65$, $p < 0.001$; spatial: $R^2 = 0.47$, $\beta = 0.302$, SE = 0.030, $t = 10.21$, $p < 0.001$) and 4 (retinal: $R^2 = 0.12$, $\beta = 0.205$, SE = 0.047, $t = 4.39$, $p < 0.001$; spatial: $R^2 = 0.02$, $\beta = 0.076$, SE = 0.049, $t = 1.53$, $p = 0.129$). Bayes Factors[63] (BF) comparing the BIC scores of these regression models indicated decisive evidence for this conclusion (Exp. 3: $\Delta$BIC $= -58.82$, BF $= 5.9 \cdot 10^{12}$; Exp. 4: $\Delta$BIC $= -15.97$, BF $= 2932$).

Similarly, movement duration was highly correlated between saccades in opposing directions (Exp. 3: $\rho = 0.822$, $p < 0.001$; Exp. 4: $\rho = 0.384$, $p < 0.001$), yet perceptual duration thresholds in both experiments showed higher correlations (side panels in Fig. 5b) with the duration of saccades sharing the same retinal (Exp. 3: $\rho = 0.849$, $p < 0.001$; Exp. 4: $\rho = 0.370$, $p < 0.001$; Fig. 5b) rather than spatial direction (Exp. 3: $\rho = 0.725$, $p < 0.001$; Exp. 4: $\rho = 0.117$, $p = 0.161$). Again, for the retinal direction in Experiment 3, these correlations remained significant even after removing the effect of saccade amplitude ($\rho = 0.41$, $p < 0.001$), while they did not for the spatial direction ($\rho = -0.05$, $p = 0.594$). Once more, saccades matched with respect to their retinal rather than their spatial direction provided the best predictor of threshold durations both in Experiment 3 (retinal: $R^2 = 0.71$, $\beta = 0.955$, $SE = 0.056$, $t = 16.99$, $p < 0.001$; spatial: $R^2 = 0.48$, $\beta = 0.784$, SE = 0.075, $t = 10.40$, $p < 0.001$; $\Delta$BIC $= -70.35$, BF $= 1.9 \cdot 10^{15}$) and 4 (retinal: $R^2 = 0.07$, $\beta = 0.439$, SE = 0.134, $t = 3.28$, $p = 0.001$; spatial: $R^2 < 0.01$, $\beta = 0.110$, $SE = 0.139$, $t = 0.79$, $p = 0.429$; $\Delta$BIC $= -9.87$, BF $= 139.2$).

Thus, both experiments provided strong evidence that individual kinematics of the retinal consequences of saccades—as opposed to the kinematics of saccades with the same direction as the stimulus—predict visibility of high-speed stimuli presented during fixation. Supplementary stepwise regressions underpin these results, showing that variability in retinal speed due to movement amplitude, direction, and individual differences each contributed uniquely to these predictions (see Supplementary Note 5).

## Main-sequence relation requires static endpoints

During natural vision, any stationary stimulus that a saccade displaces across the retina travels at high speed over a finite distance, creating static movement endpoints[10]. In the absence of pre- and post-saccadic visual input, observers perceive both motion[22] and motion smear[32–34] during saccades. Just tens of milliseconds of static input before and after the saccade eradicate the intra-saccadic percept[22,32,34]. Similarly, simulated saccadic motion of naturalistic scenes is perceived at shorter amplitudes in the presence of static movement endpoints[64]. Experiments 1 through 4 have shown that the main-sequence relation of movement speed and amplitude defines the limit of visibility of high-speed stimuli. Should the relationship to saccades hold, then reliable movement endpoint information must be instrumental for obtaining this result. To test this critical prediction, Experiment 5 manipulated the duration for which the stimulus remained stationary before and after the movement between 0, 12.5, 50, and 200 ms (static-endpoint duration), while the movement paths were identical across these conditions (as in Exp. 1).

Static-endpoint duration had a striking impact on motion visibility (expressed in terms of relative speed; Fig. 6) in that the 0 ms condition differed categorically from the others, the 12.5 ms condition differed slightly from the longer durations, and the 50 ms condition did not differ from the 200 ms condition (Helmert contrasts in Table S6 in Supplementary Note 6). Thresholds depended heavily on movement amplitude when only the motion trajectory was presented. In this 0-ms condition, psychometric functions shifted to the right with decreasing

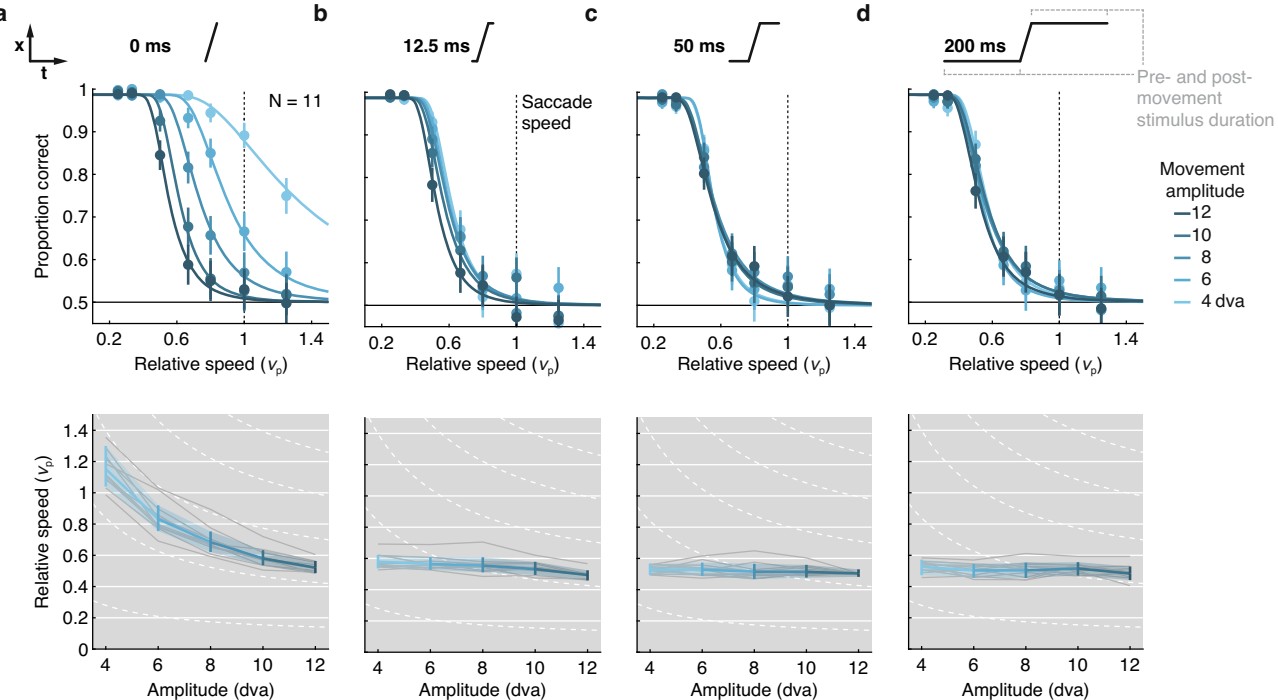

**Fig. 6 | Pre- and post-movement stimulus presence gives rise to main-sequence relation in motion visibility.** Psychometric functions and resulting visibility thresholds, expressed as relative movement speed, are shown as a function of movement amplitude and static-endpoint duration, varied between **a** 0 ms, **b** 12.5 ms, **c** 50 ms, and **d** 200 ms. Data were obtained from $N = 11$ observers completing a total of 57,672 trials. Conventions as in Fig. 3.

movement amplitude (Fig. 6a, top), and thresholds (bottom) were more consistent with the predictions of a constant speed threshold (dashed white lines) than of the main sequence (solid white lines). Accordingly, we obtained large coefficients for the linear and quadratic terms in this condition (Table S6). Surprisingly, with as little as 12.5 ms of a stationary stimulus before and after the movement, this amplitude-dependence vanished almost completely (Fig. 6b). For the 50 and 200 ms conditions (Fig. 6c, d), thresholds were independent of movement amplitude and virtually identical to those reported in Experiment 1 (where static-endpoint duration varied between 182 and 242 ms, depending on movement duration).

Thus, the key result that visibility of stimuli at high speed depends on a lawful conjunction of movement amplitude, speed, and duration (Exps. 1 through 4) is contingent on the presence of static movement endpoints (Exp. 5). Analyses of absolute movement speed, absolute movement duration, and relative movement duration were highly consistent with this conclusion (see Supplementary Note 7). These results resemble the finding that, in natural vision, stationary input before and after saccades renders intrasaccadic retinal stimulation invisible[22,32,34]. Here, the effect of endpoints appears to be twofold. On the one hand, endpoints may have served as masks[65], capable of eliminating retinal stimulus traces even across considerable spatial distances[33]. On the other hand, because localization of moving stimuli is error-prone[66], the endpoints may have served as visual references, improving access to the movement amplitude, which is essential for evaluating retinal trajectories in terms of the main sequence. A coherent account of this perceptual omission must explain how static movement endpoints give rise to the lawful relation between movement kinematics and visibility.

### Early-vision model predicts visibility

How does the conjunction of movement amplitude, speed, and duration drive visibility of high-speed stimuli?

To assess the minimal conditions under which this result could be obtained, we implemented a parsimonious model of early visual processing (Fig. 7), in which stimuli elicit neural responses in a retinotopic map of visual space upon which a decision is formed where motion was present (see "Early-vision model"). We exposed this model to the stimulus conditions used in Experiment 5, modeling discrimination performance in five steps (black number labels in Fig. 7).

First, at each presentation of the stimulus, we convolved the stimulus' current location with spatial and temporal response functions typical of early visual processing[67], yielding a neural activation map in x- and y-retinotopic coordinates across time (Fig. 7b). For slowly moving stimuli ($0.25v_p$, left panels) compared to rapidly moving stimuli ($1.00v_p$, right panels), neural activity was high during the motion portion of the stimulus, such that the stimulus' curvature (bottom panels) is highly conspicuous in the activation map. Second, we compared the maximum normalized output at a given time (Fig. 7c, solid black line) to the alternative trajectory, where motion was absent (dashed gray line). Third, the difference between these two (blue-shaded region) provides evidence that the stimulus was present along the trajectory (here, upwards). This difference had consistently larger magnitudes for slower speeds and shorter amplitudes (Fig. 7d). Fast movements created feeble visual signals, which were almost completely abolished when the stimulus had static endpoints (Fig. 7d, bottom), as compared to when endpoints were unavailable (Fig. 7d, top). Using a population aggregate (Fig. 7e), we can assess the model's estimate of stimulus position. With increasing speed, the stimulus trajectory gradually transforms from a continuous transition with curvature (top rows) to a step function with diminished curvature (bottom rows). At high speed, therefore, the stimulus gives rise to a sudden displacement (as opposed to a continuous motion) percept and, arguably, the curvature discrimination becomes impossible. Crucially this pattern only emerged with static endpoints: Both

continuous trajectories and curvature remained obvious even at the highest velocities when no endpoints were presented (0-ms rows). Fourth, the accumulated evidence, $E$, for continuous motion over all time points monotonically decreased as a function of relative movement speed (Fig. 7f). Notably, the slope of this decrease depended on amplitude for the 0 ms static-endpoint duration, but not (or much less so) if the stimulus was static at the endpoints of its trajectory for a short period of time. Finally, we compared $E$ to zero, to obtain the model's perceptual report for each simulated trial (Fig. 7g). The model's reports yielded results that were strikingly similar to those observed in human observers (Fig. 6), qualitatively reproducing the impact of pre- and post-movement stimulus presence on perception. Additional grid-search model simulations (see Supplementary Note 8) further confirmed that only physiologically plausible model parameters produced reasonable quantitative approximations of experimental results (Fig. S6).

While we based its assumptions on known temporal response properties of the primate visual system[68,69] and systematically validated our parameter selection, the model is conceptual by design and thus makes several rough simplifications. For instance, we did not model orientation or motion-selective filters[70,71], nor did we account for how spatial or temporal response properties of neurons change with stimulus eccentricity[72]. Despite its remarkable parsimony, the model captures both subjective phenomenological (Fig. 7e) and objective performance (Fig. 7g and Fig. S6) features of high-speed stimulus perception revealed by our task.

### Discussion

We discovered that the limits of high-speed finite motion perception emulate the *main sequence*–a kinematic law of oculomotor control that connects movement amplitude, peak velocity, and duration of saccadic eye movements. We had predicted this shared lawful relation between perception and eye movement kinematics based on the fundamental idea that frequent exposure of a perceptual system to the reliable sensory consequences of actions should result in adherence to these regularities even in the absence of the action itself[10–13]. In the extreme, the kinematics of actions may fundamentally constrain a sensory system's access to the physical world. Our data provide strong evidence for this idea for the case of rapid eye movements, made billions of times in a human lifetime, that each entail reliable sensory consequences: Each saccade across a visual scene results in an instantaneous, equal and opposite movement of the scene projected on the retina. The movement of the retinal image, therefore, follows the kinematic properties defined by the main sequence of saccades (Fig. 1). And as we show here, so do the limits of visibility of high-speed stimuli.

There are two key reasons why we believe that this relation is not coincidental. First, the stimulus properties governing visibility during fixation closely match those governing visibility during saccades. During fixation[45] as well as during saccades[21], humans can see motion at saccadic speeds, provided the stimulus does not have static endpoints and contains sufficiently low spatial frequencies to bring the temporal luminance modulations on the retina into a visible range. The lawful relation uncovered here holds only if the stimulus' movement is preceded and followed by tens of milliseconds of static movement endpoints (Exp. 5). Similarly, perception of intrasaccadic stimulation is omitted as soon as the scene is static for tens of milliseconds before and after the saccade[22,27,32,34]. Second, we find consistent covariations of the retinal speed imposed by observers' saccades and perception: Variations in saccade kinematics across movement amplitudes, directions, and individuals predicted corresponding variations in visibility thresholds. Importantly, these covariations were specific to the kinematics of saccades in the direction opposite of the motion stimulus. We had predicted this specificity, because the direction of retinal motion that these saccades entail matches that of the stimulus.

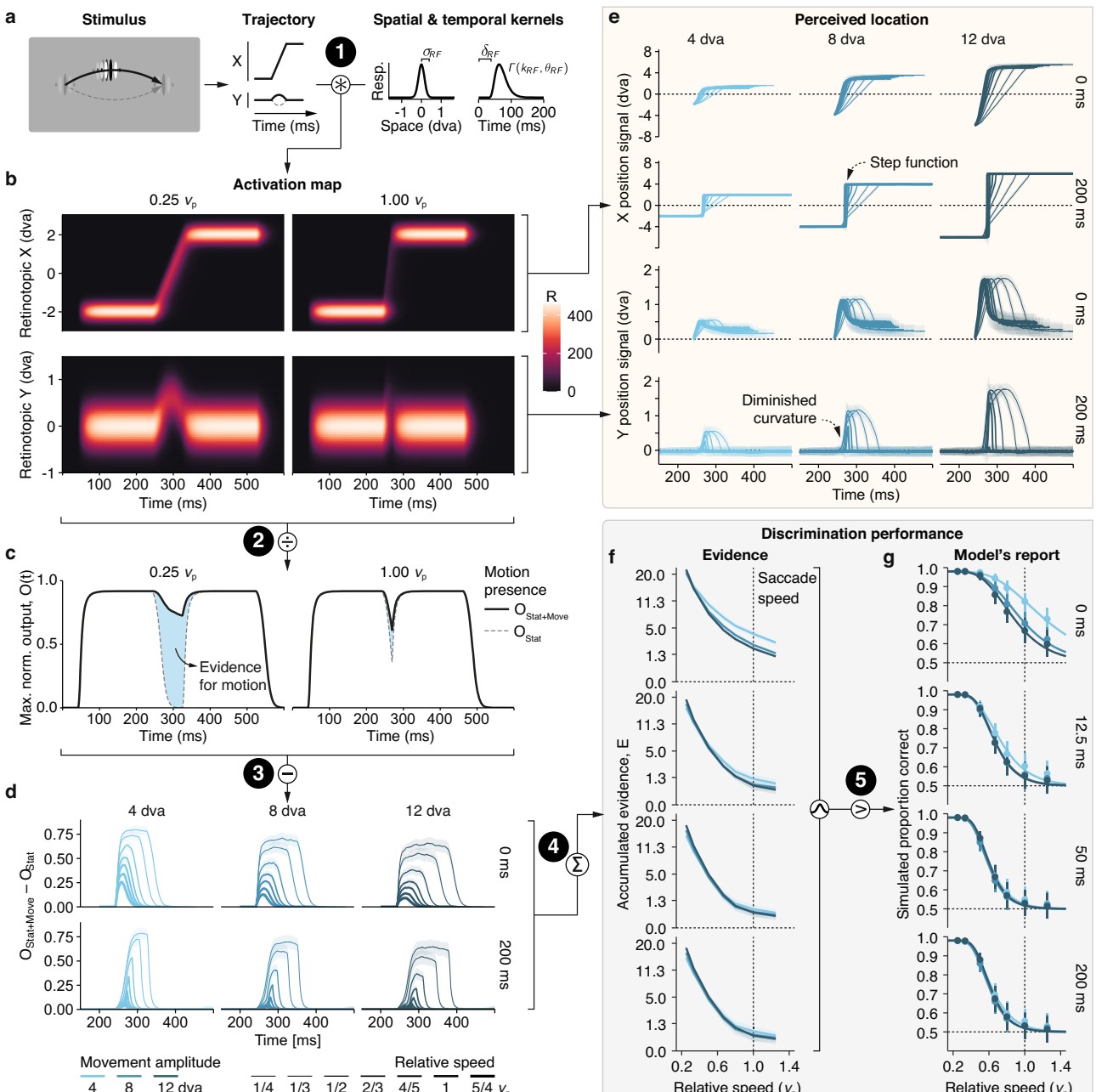

**Fig. 7 | A parsimonious early-vision model reproduces the main-sequence dependence of high-speed stimulus perception. a** We simulated stimulus trajectories introduced in Experiment 5. Convolution of the trajectory with spatial and temporal response functions yields **b** activation maps in X- and Y-retinotopic coordinates (see "Methods"). **c** The maximum normalized output of the activation map at any time point (solid black line) was compared to the output obtained when the stimulus was absent during the motion portion of the presentation (dashed gray line). **d** The resulting difference between motion-present and motion-absent as a function of movement amplitude and relative speed for 0 and 200 ms static-endpoint durations. **e** Estimate of the stimulus trajectory, read-out as a population

aggregate (see "Methods") for horizontal (top panels) and vertical (bottom panels) position for 0 and 200 ms static-endpoint durations. Data are presented as mean values ± SD across simulations. **f** Accumulated evidence, $E$, for the presence of motion as a function movement amplitude and relative speed, plotted separately for all four static-endpoint durations. **g** For each condition, samples drawn from a Gaussian distribution with $\mu = E$ and $\sigma = 4$ (see "Methods") were either above or below zero, thereby yielding correct or incorrect perceptual reports, respectively, which were then fitted analogously to the human data in Fig. 3. Error bars indicate 95% confidence intervals. White-on-black numbers highlight five steps as described in the main text.

In the current study, we have started exploring a small subset of stimulus features encountered in natural scenes. For our stimuli, visibility thresholds were a fraction of a saccade's peak velocity when the stimulus followed a constant speed (50–60% for the direction-discrimination task; 36% for the detection task in Supplementary Note 2) or a saccadic velocity profile (60–80% in Supplementary Note 3), thus covering the better part of a saccade's duration. Variations in stimulus contrast, spatial frequency, orientation, and

eccentricity will most certainly affect visibility thresholds in our task, too. While these stimulus features (and their combinations) remain to be explored empirically, we predict that they would affect thresholds overall (e.g., higher visibility for lower spatial frequencies; lower visibility for lower contrast) but in a way that retains their proportionality to the main sequence. Stimulus orientation should have a particularly strong effect on visibility thresholds, as orientations parallel to the movement direction give rise to motion streaks that are visible even

during saccades and in the presence of pre- and post-saccadic visual stimulation[27,34,73].

The fact that visibility thresholds are directly proportional to the main-sequence relations of saccades could be considered a perceptual invariance, as the visual system responds to saccadic motion consistently irrespective of differences in movement amplitude and direction. Invariances are commonplace in perception: We experience color constancy despite changes in the illumination of the environment[74], hear the same sound at the same loudness despite varying distances[75], and exploit cues in optic flow patterns during locomotion that are invariant to changes between the agent and the environment[76]. Our results point to a different type of perceptual invariance in the visual system, to self-imposed retinal motion.

How does the visual system achieve its invariance to the kinematics of saccades? We have shown that a parsimonious model of low-level visual processes combined with a simple decision-making step can qualitatively reproduce the behavioral data (Fig. 7). The fact that we see striking qualitative similarities between the simulated perceptual reports of the model and those of human observers suggests that the model captures aspects of visual processing that are key to understanding visibility of stimuli moving at high speeds. One key component of this model is the width of the temporal response function, which leads to strong and lasting responses to the static endpoints of a stimulus' trajectory. When these responses outlast the duration of a saccade, they can swallow up the weak activation resulting from motion at high speeds. However, the (presumably fixed) width of the temporal response function alone does not account for our findings. If that were the case, then absolute movement duration should predict visibility. Our data, however, show instead that with increasing movement amplitudes, longer movement durations are required to achieve the same level of visibility (Fig. 3c). Our model offers an explanation of this result, based on the relation between the activity resulting from the endpoints of the movement and the activity resulting from the movement itself. Specifically, for a given movement duration, a larger movement amplitude is associated with a higher speed than a shorter movement amplitude. The stimulus would thus spend less time in any given location, resulting in lower activation along the motion path. As a consequence, the strong activation from static endpoints can outlast longer movement durations. This speed-activation tradeoff links visibility to a conjunction of amplitude, speed, and duration of the movement. Even though our experiments up to this point were not capable of explicitly disambiguating the possible roles of endpoints as either pre- and post-movement masks or as visual cues for movement amplitude, the tight correlation of this conjunction to the main sequence relation of saccades is striking. The mechanism would thus allow for perceptual omission of intrasaccadic visual motion while maintaining high sensitivity to rapidly moving stimuli.

The correlation of high-speed visibility thresholds and saccade kinematics of individual observers (see Supplementary Note 4) confirms a tight coupling of the setup of the visual system with the kinematics of the oculomotor system that controls its input. From the outset, we hypothesized that visibility thresholds are the result of a lifetime of exposure to saccade-induced retinal motion[10], which follows the main sequence in babies after 2 months of age or earlier[77]. Indeed, the human visual system may never experience motion over finite distances at speeds higher than those imposed by saccades, such that its sensitivity is limited to that range. This direction of causality (i.e., that movement kinematics impact visual sensitivity) is consistent with recent findings that changes in the visual consequences of saccades results in quick adjustments of visual sensitivity during saccades[29]. A complementary view of our data (that would be equally adaptive) is that saccadic speeds are tuned to exploit the properties of the visual system, much like saccade amplitudes appear to be adapted to receptive field sizes and adaptive properties of neural populations in a range of different species[78]. Indeed, the kinematics of eye movements are reliable over time and across experimental conditions[56,59], and there are a number of striking examples showing that the oculomotor system resorts to keeping the kinematics of its movements relatively constant. For instance, patient HC, who could not move her eyes from birth, readily moves her head in saccade-like movements[79]. Similarly, humans whose head is slowed down by weights put on the head compensate for these external forces to regain the velocity-amplitude relation of their combined eye-head movements[80]. Finally, gaze shifts of a certain amplitude have similar dynamics, even if the eye and head movements that contribute to them have a very different composition[81]. These data suggest that the saccadic system aims to keep the kinematics of saccades constant over a large range of conditions. In the light of the data presented here, a possible function of this would be to keep movement kinematics in a range that yields perceptual omission of the saccades' retinal consequences, while maintaining a high degree of sensitivity to high-speed motion. Understanding which causal direction (or an even more complex causal structure) might underlie our observed results is a key question that follows from the lawful relation between action and perception that the present study revealed.

Irrespective of the causal direction of this mutual alignment between perception and action (which need not be unidirectional), our results have intriguing consequences for mainstream theories that rely on corollary discharge signals to explain perceptual experience. Corollary discharge signals exist across a wide range of species[82]. They support key functions in motor control and spatial updating[83] and contribute to visual stability[36,84] and attentional continuity across saccades[85–87]. Moreover, disturbed corollary discharge provides an explanation for psychotic symptoms such as disruptions in agency[88]. Corollary discharge has also been used to explain various forms of sensory attenuation during goal-directed movements, including reduced contrast sensitivity during saccades[42–44], an idea that dates back to Helmholtz's reafference principle[89]. Relying on corollary discharge, however, requires tightly-timed, long-range communication between motor and sensory areas of the brain, and an expeditious translation of motor signals into sensory predictions. Our results suggest a simpler alternative based only on reafferent signals that uniquely characterize an action: The lawful kinematics of an actions' sensory consequences by themselves might give rise to perceptual omission. Experimental evidence that predictions based on sensorimotor contingencies play a role in saccadic omission has recently been presented by demonstrating that the strength of intra-saccadic motion percepts could be downregulated after habituation[90]. Because our proposed mechanism does not strictly require the idea of a corollary-discharge-based prediction, it is a parsimonious explanation of the current results that were obtained during fixation in the absence of saccades, as well as saccadic omission in natural vision. It does not, however, constitute an explanation for the reduction of contrast sensitivity for briefly flashed gratings, which are widely used to characterize contrast sensitivity during saccades[42–44]. Recent neurophysiological evidence suggests, however, that these effects may be caused by visual mechanisms as well: Contrast sensitivity reduction can be observed in retinal ganglion and bipolar cells, thus clearly in the absence of influences from corollary discharge[39]. While this does not exclude an impact of extra-retinal signals in saccadic omission, their role might be different than previously assumed. For instance, they could enhance contrast sensitivity upon saccade landing[39], or they might be part of a sensorimotor contingency that is used to down-regulate the visual consequences of eye movements[29]. Delineating the unique contributions of various signals to saccadic omission constitutes an exciting line of research that, of course, comes with severe methodological challenges (e.g., to stabilize the whole-field visual input during saccades; to faithfully replay the visual consequences of saccades during fixation; to equate the deployment of visual attention across saccade and fixation conditions; etc.)[10]. However, compared to a simple reduction in visual

sensitivity (whether of retinal or extra-retinal origin), a mechanism based on the kinematics of saccades seems appealing: It would allow for the perceptual omission of motion during saccades while maintaining maximal sensitivity to high-speed motion during fixation (with a boundary defined by saccade kinematics). This retained sensitivity comes with at least two benefits: First, the visual system would miss little (potentially relevant) motion in the world while largely ignoring motion caused by one's own eye movements. And, second, residual sensitivity to the slower phases of saccade-induced motion would bleed through[34], which may help tracking objects' swiftly changing positions on the retina when the eyes move[28].

Our results reveal a lawful relation between action kinematics and the limits of human visual perception. Future research should investigate if such coupling generalizes across species (e.g., faster-moving animals should be more sensitive to high-speed motion than slower-moving animals) and sensory modalities (e.g., auditory motion perception may be constrained by the kinematics of head movements), in particular, if the actions sampling the environment impose regularities onto the input of the sensory system. The present study suggests that the functional and implementational properties of a sensory system are best understood in the context of the kinematics of actions that drive its input.

## Methods
### Participants
For all experiments, we recruited participants through word of mouth and campus mailing lists. They were naïve as to the purpose of the study, had normal or corrected-to-normal vision, and received monetary compensation for their participation. Before starting the first session, they provided informed written consent. Nobody besides a research assistant and the participant were present during the data collection. Experimenters were blinded to the experimental conditions and the study hypotheses. All studies were done in agreement with the Declaration of Helsinki in its latest version (2013), approved by the Ethics Committee of the Deutsche Gesellschaft für Psychologie (Experiments 1, 3, and 4) or the Ethics board of the Department of Psychology at Humboldt-Universität zu Berlin (Experiments 2a, 2b, and 5). No statistical method was used to predetermine sample size. All experiments were pre-registered at the Open Science Framework (OSF; links provided below). Participant gender data summarized below was based on self-report. The variable was surveyed for these descriptive purposes only and considered in neither study design nor data analyses.

**Experiment 1.** Ten participants (18–32 years old; 7 female; 6 right-eye dominant; all right-handed) completed all sessions and received 25€ as remuneration. Three additional participants were recruited but had to be excluded as they did not complete all sessions. The pre-registration of the study is available at https://doi.org/10.17605/OSF.IO/AJY7D.

**Experiment 2a.** Ten participants (19–28 years old; 8 female and 2 male; 3 right-eye dominant; 8 right-handed) completed all sessions and received 44€ as remuneration. Three additional participants were recruited but had to be excluded as they did not complete all sessions. The pre-registration of the study is available at https://doi.org/10.17605/OSF.IO/32QFP.

**Experiment 2b.** Ten participants (21–35 years old; 6 female and 4 male; 8 right-eye dominant; 9 right-handed) completed all sessions and received 40€ as remuneration. No participants had to be excluded. The pre-registration of the study is available at https://doi.org/10.17605/OSF.IO/WUKRC.

**Experiment 3.** Six participants (19–32 years old; 5 female and 1 male; 3 right-eye dominant; all right-handed) completed all sessions and

received 52€ as remuneration. One additional participant was recruited but had to be excluded as they did not complete all sessions. The pre-registration of the study is available at https://doi.org/10.17605/OSF.IO/dwvj2.

**Experiment 4.** 40 participants (19–40 years old; 30 female and 10 male; 27 right-eye dominant; 35 right-handed) completed one session each and received 10€ as remuneration. Four participants had to be excluded because of incomplete data sets. One person could only complete 75% of their session and was included. The final sample consisted of 36 participants (19–40 years old years old; 28 female; 25 right-eye dominant; 33 right-handed). The pre-registration of the study is available at https://doi.org/10.17605/OSF.IO/s6dvb.

**Experiment 5.** Ten participants (19–38 years old; 8 female and 2 male; 7 right-eye dominant; 7 right-handed) completed all sessions and received 44€ as remuneration. Two additional participants were recruited but had to be excluded; one did not complete all sessions and one performed at chance level. The pre-registration of the study is available at https://doi.org/10.17605/OSF.IO/7nv5f.

### Apparatus
Stimuli were projected onto a standard 16:9 (200 × 113 cm) video-projection screen (Celexon HomeCinema, Tharston, Norwich, UK), mounted on a wall, 270 cm (Experiments 1, 3, and 4) and 180 cm (Experiments 2a, 2b, and 5) in front of the participant, who rested their head on a chin rest. The high-speed PROPixx DLP projector (Vpixx Technologies, Saint-Bruno, QC, Canada) updated the visual display at 1440 Hz, with a spatial resolution of 960 × 540 pixels. The experimental code was implemented in MATLAB (Mathworks, Natick, MA, USA), using the Psychophysics and Eyelink toolboxes[91,92] running on a Dell Precision T7810 Workstation with a Debian 8 operating system. Eye movements were recorded via an EyeLink 2 head-mounted system (SR Research, Osgoode, ON, Canada) at a sampling rate of 500 Hz, except in Experiments 2a, 2b, and 5, in which we used an EyeLink 1000+ system at a sampling rate of 1000 Hz. Responses were collected with a standard keyboard.

### Procedure in perceptual trials (Experiments 1–5)
Trials probing stimulus visibility had the same general task and procedure across all experiments; exceptions are detailed in Experimental variations. Each trial was preceded by a fixation check for which a fixation spot (diameter: 0.15 degrees of visual angle, dva) was displayed at the center of the screen. Once fixation was detected in an area of 2.0 dva around the fixation spot for at least 200 ms, the trial started. The fixation spot remained on the screen throughout the trial. A Gabor stimulus appeared either left or right of the screen center (chosen randomly on each trial), ramping up from zero to full contrast in 100 ms. At full contrast, the stimulus rapidly moved—in a curved trajectory such that it passed the center above or below the fixation spot—towards the other side of fixation, and stopped before ramping back to zero contrast in 100 ms. Altogether the stimulus was on the screen for 500 ms with its motion centered during that time period (i.e., it started earlier and ended later for longer motion durations). Once the stimulus had disappeared, the observer pressed one of two buttons to indicate whether the stimulus moved in an upward or a downward curvature.

Stimuli were vertically-oriented Gabor patches (1 cycle/dva, sigma of the envelope: 1/3 dva), traveling on a motion path corresponding to an arc of a circle with a radius chosen such that the maximum deviation from a straight line was exactly 15% (reached at the center of the screen, right above or below fixation). The amplitude of the movement, $A$ was either 4, 6, 8, 10, or 12 dva. The stimulus' horizontal velocity remained constant throughout its motion, and proportional to the peak velocity, $v_p$, of a horizontal saccade at that amplitude as

**Table 1 | Stimulus parameters**

| $v_{rel}$ | $D_{rel}$ | | Amplitude (dva) | | | | |
|---|---|---|---|---|---|---|---|
| | | | 4 | 6 | 8 | 10 | 12 |
| 1/4 | | $v$ | 48.5 | 62.8 | 73.5 | 82.0 | 88.8 |
| | 2.44 | $D$ | 82.4 | 95.6 | 108.8 | 122.0 | 135.1 |
| 1/3 | | $v$ | 64.7 | 83.7 | 98.1 | 109.3 | 118.4 |
| | 1.83 | $D$ | 61.8 | 71.7 | 81.6 | 91.5 | 101.3 |
| 1/2 | | $v$ | 97.0 | 125.5 | 147.1 | 164.0 | 177.6 |
| | 1.22 | $D$ | 41.2 | 47.8 | 54.4 | 61.0 | 67.6 |
| 2/3 | | $v$ | 129.4 | 167.3 | 196.1 | 218.7 | 236.8 |
| | 0.91 | $D$ | 30.9 | 35.9 | 40.8 | 45.7 | 50.7 |
| 4/5 | | $v$ | 155.3 | 200.8 | 235.3 | 262.4 | 284.2 |
| | 0.76 | $D$ | 25.8 | 29.9 | 34.0 | 38.1 | 42.2 |
| 1 | | $v$ | 194.1 | 251.0 | 294.2 | 328.0 | 355.2 |
| | 0.61 | $D$ | 20.6 | 23.9 | 27.2 | 30.5 | 33.8 |
| 5/4 | | $v$ | 242.6 | 313.8 | 367.7 | 410.0 | 444.0 |
| | 0.49 | $D$ | 16.5 | 19.1 | 21.8 | 24.4 | 27.0 |

Absolute movement speeds ($v$ in dva/s) as well as absolute ($D$ in ms) and relative ($D_{rel}$ in units of $D_{sac}$) movement durations as a function of the experimentally manipulated relative movement speeds ($v_{rel}$ in units of $v_p$).

described the main-sequence relation[18], $v_p = c \cdot A/D_{sac}$, where $c$ is a dimensionless proportionality constant of 1.64, and $D_{sac}$ is the average duration of a horizontal saccade, as captured by its linear relation to saccade amplitude[19], $D_{sac} = 2.7 \cdot A + 23$ ms. We varied absolute stimulus speeds ($v$) for each amplitude by multiplying the corresponding $v_p$ by relative movement speeds, $v_{rel} = \{1/4, 1/3, 1/2, 2/3, 4/5, 1, 5/4\}$. The duration of movements at these velocities correspond to $D = A/(v_{rel} \cdot v_p)$. The resulting speeds and durations used in the experiments are displayed in Table 1. We verified the timing of stimuli at high velocities with photodiode measurements (see Supplementary Methods: Faithful rendering of high-speed motion).

Online fixation control ensured that participants fixated throughout the trial. Trials in which fixation was not maintained during stimulus presentation were aborted and repeated in random order at the end of the block.

**Procedure in saccade trials (Exp. 3 and 4)**
Saccade trials started with a fixation check at a location offset from the center of the screen in the horizontal or vertical direction. Once fixation was detected in an area of 2.0 dva around the fixation spot (black; diameter: 0.15 dva) for at least 200 ms, the trial started. The fixation spot then jumped to the opposite side of the screen center, and participants executed a saccade to its new location. Initial and end points of the fixation target were offset from the screen center by half the target's eccentricity (i.e., instructed saccade amplitude). We detected the execution of saccades online, by registering saccade landing within a radius of 50% of the target eccentricity within 400 ms of target onset. Target eccentricity varied between 4 and 12 dva, in steps of 1 dva (randomly interleaved across trials).

**Experimental variations**
**Experiment 1.** Participants completed three sessions of data collection, each consisting of 1400 perceptual trials, distributed over 10 blocks (140 trials per block). Each combination of movement amplitude, movement velocity, curvature direction (up or down), and motion direction (leftward or rightward), occurred once per block of trials; all combinations were randomly interleaved. For all analyses, we collapsed across curvature directions and motion direction, resulting in a total of 120 trials per data point.

**Experiment 2a.** Participants completed five sessions of data collection, each consisting of 1400 perceptual trials, distributed over 10 blocks (140 trials per block). Each combination of movement amplitude, movement velocity, presence of continuous motion (present or absent), and motion direction (leftward or rightward) occurred once per block of trials; all combinations were randomly interleaved. For all analyses, we collapsed across motion direction, resulting in a total of 200 trials per data point (including 100 present and 100 absent trials). In contrast to all other experiments reported, the motion path of the Gabor did not have a vertical component and, thus, no curvature. As the Gabor would pass directly through fixation, the fixation dot disappeared for the duration that the stimulus was on the screen. We manipulated the presence (50% of the trials) vs absence (50% of the trials) of continuous motion, and asked participants to detect the presence of motion by pressing one of two buttons (present vs absent). In trials with no continuous motion, the Gabor disappeared for the duration that the stimulus would have moved in the continuous motion condition, creating separate motion-absent conditions for each combination of amplitude and speed.

**Experiment 2b.** Experiment 2b was identical to Experiment 2a, with the following exceptions. Participants completed four sessions of data collection, each consisting of 1050 perceptual trials, distributed over 15 blocks (70 trials per block). Each combination of movement amplitude, movement velocity, and presence of continuous motion (present or absent) occurred once per block of trials; all combinations were randomly interleaved. The range of relative movement speeds was increased to $v_{rel} = \{1/4, 1/(2\sqrt{2}), 1/2, 1/\sqrt{2}, 1, \sqrt{2}, 2\}$, to account for the higher performance that we expected from a time-varying velocity profile. For all analyses, we collapsed across motion direction (leftward or rightward), which was randomly drawn on each trial, resulting in a total of 120 trials per data point (including 60 present and 60 absent trials).

In trials with continuous motion, the horizontal velocity of the stimulus followed a speed profile that mimicked the velocity profile of saccades, following the profile of a gamma function[18], including a rapid acceleration phase and a slower deceleration phase (Fig. S3a):

$$v(t) = \alpha \left[\frac{t}{\beta}\right]^{\gamma-1} \cdot \exp\left[-\frac{t}{\beta}\right] \tag{1}$$

where $v(t)$ is the saccadic velocity profile. $\alpha$ and $\beta$ are scaling constants and $\gamma$ is a shape parameter that determines the asymmetry of the velocity profile. While gamma depends on movement amplitude and was determined according to published parameters ($\gamma = 4/(3.26D+0.77)^2$)[18], $\alpha$ and $\beta$ were scaled to accommodate the desired relative movement speed (see Fig. S3a). Note that the simulated movements did not include post-saccadic oscillations, which are able to introduce retinal image shifts of ~0.033 dva per degree of saccade amplitude[93,94] (due to inertial motion of the crystalline lens), but also entail measurable perceptual consequences[34,95]. Compared to real saccades, our simulated saccade profiles—assuming our largest movement amplitude (i.e., 12 dva) and a corresponding lens overshoot of 4 dva[93]—may thus have underestimated the target stimulus' post-saccadic movement amplitude by up to 0.4 dva. Due to the difficulty involved in modeling the nonlinear dynamics of post-saccadic oscillations and, more critically, their retinal consequences[61,62], and because our stimulus endpoints were in peripheral rather than foveal locations, we chose not to include this aspect in the stimulus trajectories of Experiment 2b.

In trials with no continuous motion, the Gabor underwent a quick contrast ramp for the duration that the stimulus would have moved in the continuous motion condition, creating separate motion-absent

conditions for each combination of amplitude and speed. The contrast ramp followed the mirrored profile of the saccadic velocity profile, $c(t) = 1 - v(t)/\max v$, going from full contrast to zero contrast and back to full contrast. The stimulus jumped from its initial to its final position when contrast was exactly zero (corresponding to the moment of peak velocity in the continuous motion condition). In all trials, the pre- and post-movement stimulus duration was fixed at 100 ms; the stimulus had 100% contrast when it first appeared and just before it disappeared (i.e., there was no contrast ramping at the beginning and end of the trial).

**Experiment 3.** Participants completed six sessions of data collection, each consisting of 1408 trials, distributed over a total of 8 blocks, alternating between blocks of perceptual trials (280 trials per block; 4 blocks per session) and saccade trials (72 trials per block; 4 blocks per session). In perceptual trials, motion direction of the Gabor stimulus was either horizontal (from left to right or vice versa) or vertical (from top to bottom or vice versa); motion curvature was orthogonal to the motion direction. Each combination of movement amplitude, movement velocity, curvature direction (up vs down or left vs right), and motion direction (leftward vs rightward vs upward vs downward), occurred once (total number of perceptual blocks: 24). For all analyses, we collapsed across curvature directions, resulting in a total of 48 trials per data point. In saccade trials, each combination of target eccentricity and saccade direction was tested twice in each block of 72 saccade trials (total number of saccade blocks: 24).

**Experiment 4.** Participants completed one session of data collection, consisting of 1408 trials, distributed over a total of 8 blocks, alternating between blocks of perceptual trials (280 trials per block; 4 blocks per session) and saccade trials (72 trials per block; 4 blocks per session). In perceptual trials, motion direction of the Gabor stimulus was either horizontal (from left to right or vice versa) or vertical (from top to bottom or vice versa); motion curvature was orthogonal to the motion direction. We tested only the 8 dva movement amplitude. Each combination of movement velocity, curvature direction (up vs down for horizontal, or left vs right for vertical motion), and motion direction (leftward vs rightward vs upward vs downward), occurred five times per block of trials. For all analyses, we collapsed across curvature directions, resulting in a total of 40 trials per data point. In saccade trials, each combination of target eccentricity and saccade direction was tested twice in each block of 72 saccade trials.

**Experiment 5.** Participants completed five sessions of data collection, each consisting of 1120 perceptual trials, distributed over 8 blocks (140 trials per block). Each combination of movement amplitude, movement velocity, static-endpoint duration, curvature direction (up or down), and motion direction (leftward or rightward), occurred once per block of trials; all combinations were randomly interleaved. For all analyses, we collapsed across curvature directions and motion direction, resulting in a total of 40 trials per data point. The motion of the Gabor stimulus was identical to that in the previous experiments, but we varied the pre- and post-movement stimulus duration between 0, 12.5, 50, or 200 ms. The stimulus had 100% contrast for as long as it was present (no contrast ramping).

## Data pre-processing
We detected saccades based on their 2D velocity[96]. Specifically, we computed smoothed eye velocities using a moving average over five subsequent eye position samples in a trial. Saccades exceeded the median velocity by 5 SDs for at least 8 ms. We merged events separated by 10 ms or less into a single saccade, as the algorithm often detects two saccades when the saccade overshoots at first.

In perceptual trials, we confirmed successful fixation during each trial offline. Trials with saccades larger than 1 dva during stimulus presentation were excluded, as were trials with missing data (e.g., due to blinks) or skipped frames. These criteria resulted in the exclusion of 1,695 (4.1% of 41,440) in Experiment 1, 1017 (1.5% of 70,006) in Experiment 2a, 1044 (2.5% of 41,981) in Experiment 2b, 1663 (4.1% of 40,319) in Experiment 3, 1485 (3.7% of 40,037) in Experiment 4, and 647 (1.2% of 56,140) in Experiment 5 perceptual trials from subsequent analyses, respectively.

In saccade trials, we defined response saccades as the first saccade leaving a fixation region (radius: 2 dva) around initial fixation and landing inside an area around the saccade target (radius: half the target eccentricity). Trials with saccades larger than 1 dva prior to the response saccade were discarded, as were trials with missing data (e.g., due to blinks) or saccadic gains (amplitude/eccentricity) smaller than 0.5 or larger than 1.5. After pre-processing, a total of 471 (4.5% of 10,368) and 1031 trials (10.0% of 10,296) were rejected based on these criteria and not included in subsequent analyses of Experiments 3 and 4, respectively.

## Analysis of psychophysical data
We assessed performance (visibility during high-speed motion) by computing observers' percentage of correct identification of the stimulus' curvature (up or down) in each stimulus condition (e.g., a combination of movement amplitude and movement speed). Using hierarchical Bayesian modeling with JAGS[97] Markov-Chain-Monte-Carlo sampling (5000 chains) in the Palamedes toolbox[50] (Version 1.11.2), we then fitted sigmoidal psychometric functions to the performance values of a given stimulus parameter (i.e., a set of absolute movement speeds in Fig. 3a). We used Gumbel functions defined as

$$\Psi(x; \theta, \beta, \gamma, \lambda) = \gamma + (1 - \gamma - \lambda)(1 - e^{-10^{\beta(x-\theta)}}) \qquad (2)$$

where $\Psi(x)$ is the proportion correct at a stimulus value $x$, $\theta$ is the visibility threshold, $\beta$ is the slope of the function, $\gamma$ is the guess rate, and $\lambda$ is the lapse rate. Before fitting, stimulus parameters $x$ (i.e., movement speeds or durations) were log10-transformed, as required when using Gumbel functions. For negative-slope psychometric functions (i.e., for absolute speed, relative speed, and relative duration), the sign of these log10-transformed variables was inverted before fitting and then inverted back for reporting and plotting. Hierarchical Bayesian modeling allowed us to fit a single model that simultaneously captured parameter estimates of psychometric functions for all conditions and individual observers that were tested in a given experiment[50]. In each model, $\theta$ and $\beta$ were free to vary across observers and conditions. $\gamma$ was fixed at 0.5 and $\lambda$ was constrained to be the same across all conditions within each observer, but were allowed to vary across observers. For Gumbel functions, thresholds evaluate to $1 - e^{-1} = 63.21\%$ of the function's range. Thus, $\theta$ is the value at which an observer's proportion correct equals

$$\Psi(\theta) = \gamma + (1 - e^{-1})(\gamma - \lambda) \qquad (3)$$

which corresponds to 81.6% correct for a two-alternative-forced-choice task and lapse rate of zero.

For inferential statistics, we reparameterized threshold parameters in each model as orthogonal contrasts[98,99] that evaluated the impact of a stimulus variable on visibility thresholds. Essentially, contrasts represented weighted sums of the thresholds estimated in each condition (i.e., linear combinations) allowing to evaluate specific hypotheses. Depending on the question addressed, we used Helmert contrasts, in which each level of a stimulus variable is compared to the mean of the subsequent levels, or Polynomial contrasts, which test if thresholds had linear, quadratic, or higher-order trends as a function of a stimulus variable. For descriptive statistics, we obtained the mode

of the posterior density as the central tendency of each parameter and the 95% highest-posterior-density intervals as credible intervals (CIs).

Using Bayesian hierarchical modeling to fit psychometric functions may cause shrinkage of the estimated visibility thresholds (i.e., more extreme individual thresholds will shift towards the mean). We thus used rank correlations (Spearman's $\rho$) to relate visibility thresholds to saccadic parameters.

## Analysis of saccade kinematics

When using video-based eye tracking, raw peak velocity measurements overestimate the true velocity of the eye, due to inertial forces acting upon elastic components such as the iris and the lens, with respect to the eyeball[62,100,101]. When using video-based eye tracking, it is specifically the pupil within the iris that moves relative to the corneal reflection[102]. To correct for these distortions, we fitted a biophysical model[61,103] to each observer's individual saccade trajectories, recorded with the video-based Eyelink 2 system, to estimate the physical velocity of the eyeball[62].

Specifically, we first extracted raw gaze trajectories for both left and right eye for each saccade detected in Experiments 3 and 4. To prepare the data for fitting, we normalized these trajectories such that they could be interpreted as distance traveled over time relative to the time and position of saccade onset. In all fitted biophysical models, we assumed constant elasticity and viscosity parameters $\gamma$ and $k$ for each observer, and fixed the forcing parameters $\beta$ to 1 and $x_m$ to the saccade's amplitude, reducing computational complexity. We first fitted models separately for left and right eye, to each individual trajectory according to a previously described procedure[62]: Starting parameters were determined in a grid search ($\mu = [1.5, 3]$, $A = [0.01, 0.09]$, $\gamma_0 = [0.05, 0.7]$, $k_0 = [0.01, 0.14]$) and model optimization, using the Levenberg-Marquart algorithm, was performed starting from grid-search results. Based on the resulting parameter estimates, we computed the underlying eyeball trajectory[103]. Second, peak velocity and duration were extracted from the eyeball's trajectory. Specifically, saccadic peak velocity was defined as the maximum sample-to-sample velocity present in the estimated eyeball trajectory, and saccade duration was computed based on a threshold defined by the median-based standard deviation of the same sample-to-sample velocity. Peak velocities larger than 800 deg/s (0.7% of all trajectories) and durations longer than 100 ms (0.9% of all trajectories) were excluded from further analysis. Third, to estimate main-sequence relationships, we fitted two functions[19], one predicting peak velocity and another to predict saccade duration based on saccade amplitude, to the combined data of left and right eye, separately for all four cardinal saccade directions. To predict peak velocity $v_p$, we fitted the function

$$v_p = v_0 \left( 1 - \exp\left( \frac{-A}{a_0} \right) \right) \tag{4}$$

and, to predict saccade duration, we fitted the linear relationship

$$D = d_0 + c \cdot A \tag{5}$$

where $A$ denotes saccade amplitude in dva. Fits were performed in a mixed-effects framework using the R package *nlme*[104] (starting parameters: $v_0 = 37.5$, $a_0 = 0.66$, $d_0 = 1.5$, $c = 0.18$). Models included observers as random effects, thus allowing individual parameter estimates for each observer and saccade direction condition.

## Early-vision model

To simulate visual processing of the stimulus used in our experiments, we implemented a model that convolved retinotopic stimulus trajectories over time (and a size corresponding to the aperture width of $\sigma_{stim} = 1/3$ dva we used in our experiments) with spatial and temporal response functions characteristic of the early visual system. Spatial

processing (see Fig. 7b) was modeled using two-dimensional Gaussian kernels with a standard deviation of $\sigma_{RF} = 0.15$ dva[67], whereas temporal response functions were approximated by a modified Gamma distribution function[34] with an arbitrary latency of $\delta_{RF} = 40$ ms, and shape and scale parameters set to $k_{RF} = 1.6$ and $\theta_{RF} = 12.5$, respectively. Note that, as the crucial aspect of this model lies in the temporal dynamics of processing, we did not take into account orientation selectivity, but instead chose to simplify the vertically oriented Gabor patch to a Gaussian blob. Consequently, no motion-selective mechanisms were implemented either, as motion processing was also not critical to perform the experimental task. The convolution with spatial and temporal kernels resulted in the space-time-resolved visual activity in response to the stimulus $R(x, y, t)$. Spatial and temporal resolution was set to 0.05 dva and 0.69 ms, respectively. Activity was then normalized using the Naka-Rushton transformation to compute output $O$ [cf. ref. 42, Eqs. 4–5]

$$O(x, y, t) = \frac{R(x, y, t)^2}{R(x, y, t)^2 + C^2} \tag{6}$$

where $C$ is the value at which this hyperbolic function reaches 50%. $C$ was (arbitrarily) defined as 30% of the maximum visual activity found across all conditions. Subsequently, as visual information anywhere in retinotopic space could be used as evidence for the presence and location of the stimulus, the output was reduced to

$$O(t) = \max O(x, y, t) \tag{7}$$

that is, the maximum output across the visual field at each time point. Relying on probability summation[42,105], evidence for the presence of continuous stimulus motion $E$, defined as the summed output difference between the stimulus' trajectory and its alternative (that did not contain motion), could be accumulated according to

$$E = \left( \sum_t |O_{Stat+Move}(t) - O_{Stat}(t)|^\beta \right)^{\frac{1}{\beta}} \tag{8}$$

where the accumulation slope $\beta$ was set to 1.5, $O_{Stat+Move}$ denotes output from the trajectory with continuous motion (and endpoints) present, and $O_{Stat}$ denotes output from the alternative trajectory where motion was absent (endpoints-only). If output across these two trajectories were identical over time, then evidence would be zero.

Finally, to convert evidence to discrete responses, we used a simple sampling procedure: From 200 sampled values (i.e., trials), we computed proportions of correct perceptual judgments, taking into account an arbitrary lapse rate of $\lambda = 0.02$. Correct responses $X$ were sampled according to

$$X = \begin{cases} 1, & \text{if } \mathcal{N}(E, \sigma_E) > 0 \\ 0, & \text{if } \mathcal{N}(E, \sigma_E) \leq 0 \end{cases} \tag{9}$$

where $\mathcal{N}$ is the normal distribution, $E$ is the evidence in each condition, and $\sigma_E = 4$. This procedure was repeated a thousand times for each condition to estimate the confidence intervals shown in Fig. 7g, defined by the 0.025 and 0.975 quantiles of each resulting distribution. We fitted Weibull functions to the proportions of correct responses as a function of (linear) relative-speed, and fixed guess and lapse rates ($\gamma = 0.5$ and $\lambda = 0.02$).

The model was also designed to output a stimulus position signal over time. To elaborate, due to the sluggishness of the temporal response functions found in the visual system[68,69], the representation of stimulus location, given by the population response across many receptive fields, need not necessarily be veridical. In fact, we showed that the model predicts the phenomenological appearance of the

target's trajectory as it turns from continuous motion into step-like motion and gradually reduces visible curvature as target speed increases Fig. 7e). While there are sophisticated ways of decoding population responses[106], our model's position estimate at each time point $\tau$, $\tau \in t$, was defined based on the arithmetic means of $x$ and $y$ coordinates, each weighted by the product of visual output weights $w_O$ and distance weights $w_\Delta$. Output weights were defined as

$$w_O(x,y) = \frac{O(x,y,\tau)}{\max O(x,y,\tau)} \quad (10)$$

and distance weights were defined as

$$w_\Delta(x,y) = \exp\left(-\frac{1}{2}\left(\frac{\sqrt{(x-x_{max})^2+(y-y_{max})^2}}{\sigma_I}\right)^2\right) \quad (11)$$

where $x_{max}$ and $y_{max}$ are the retinotopic coordinates with the maximum output $\max O(x,y,\tau)$, and $\sigma_I$ is the standard deviation of the Gaussian integration window, set to 2 dva. While output weights biased position estimates towards those coordinates with high output, distance weights restricted the spatial range across which coordinates could be integrated. For instance, in the presence of two distinct high-output hotspots rather far away from each other, distance weights would ensure that the resulting position estimate would not be a mere (illogical) average of the two, but that the hotspot with higher output would determine the position signal over the other.

To introduce uncertainty in the model, two sources of noise were introduced in the process of visual processing. First, we simulated ocular drift using a self-avoiding random walk model[107], using a lattice size of 2 dva, a relaxation rate of 0.001, and a 2D quadratic potential with a steepness of 1. We used a modification of this original model which smoothed position samples with a five-point running mean to avoid discreteness of resulting steps and reduce velocity noise[62]. Second, we added amplitude-dependent noise to each visual response $R$ according to the Gaussian distribution $\mathcal{N}(0, \frac{1}{8}R)$, as variance of neuronal responses has been shown to increase with their amplitude[108]. Finally, model simulations were run 250 times per condition, introducing noise at each run, independently for the Stat+Move vs the Stat trajectories.

### Reporting summary
Further information on research design is available in the Nature Portfolio Reporting Summary linked to this article.

## Data availability
Both compiled and raw eye-tracking data of all experiments is made publicly available at https://doi.org/10.17605/OSF.IO/QY4DC in csv and text format, respectively. Eye-tracking data used for the analysis of saccade kinematics for Experiments 3 and 4 can be found at https://doi.org/10.17605/OSF.IO/AVRPX.

## Code availability
Analysis code is deposited alongside experimental data at https://doi.org/10.17605/OSF.IO/QY4DC. Code for the analysis of saccade kinematics is provided at https://doi.org/10.17605/OSF.IO/AVRPX. All code relevant to modeling is also publicly available and can be found at https://doi.org/10.5281/zenodo.14745417.

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

## Acknowledgements

We thank Nina Hanning, Lisa M. Kroell, Matthias Nau, Oliver Steiner, and Viola Störmer for feedback on an earlier version of this manuscript, and all members of the Active Perception and Cognition laboratory for their continuous contribution to the project, including help with data collection. M.R. is grateful to Patrick Cavanagh, Casimir Ludwig, Michael N. Shadlen, Mohammad Shams, and Mark Wexler for insightful discussions about the results, and to Nicolaas Prins for advice on statistical contrasts in Palamedes. This research was supported by the European Research Council (ERC) under the European Union's Horizon 2020 research and innovation program (grant No 865715) as well as the Emmy Noether and Heisenberg Programmes of the Deutsche Forschungsgemeinschaft, DFG (grants RO 3579/2-1, RO 3579/8-1, RO 3579/10-1 and RO 3579/12-1). M.R. and R.S. were supported by the DFG under Germany's Excellence Strategy—EXC 2002/1 "Science of Intelligence"—project no. 390523135. R.S. was funded by Studienstiftung des deutschen Volkes during the early stages of the project. M.R. wrote the first draft of the manuscript during a sabbatical at Dartmouth College, funded by the Harris German/Dartmouth Distinguished Visiting Professorship program. The collaboration between M.R. and T.W. was supported by a DAAD-AU grant. We acknowledge support by the Open Access Publication Fund of Humboldt-Universität zu Berlin.

## Author contributions

M.R., R.S., T.L.W., and S.O. conceived the idea. M.R., R.S., and S.O. designed the experiments. M.R. and R.S. performed the research and analyzed the data. R.S. devised and performed modeling. M.R. wrote the initial draft of the paper, and R.S. contributed individual subsections. M.R., R.S., E.C., T.L.W., and S.O. revised, edited, and commented the draft and approved the final version. E.C. provided the conceptual mentorship that led to this project in the first place.

## Funding

## Competing interests

The authors declare no competing interests.
