## [Transparent Peer Review file · Nature Communications]

Lawful kinematics link eye movements to the limits of high-speed perception

Corresponding Author: Professor Martin Rolfs

Version 0:

Reviewer comments:

Reviewer #1

(Remarks to the Author)

This manuscript describes a set of experiments examining the relationship between the limits of visibility for fast moving stimuli and regularities in the retinal motion produced by saccadic eye movements. Upper-speed thresholds for detecting curvature in the path of translating Gabor patches are shown to scale with the spatial extent of motion, in a manner that is predictable consistent with the main sequence for saccades. Speed thresholds are also shown to systematically covary with saccade kinematics across individuals and manipulations of direction/amplitude. A model of early visual processing incorporating spatial and temporal blur filters is developed that reproduces the scaling of speed thresholds with spatial extent of motion, along with the finding that this relationship only holds when the stimulus remains visible at the start & endpoints of the trajectory. The authors argue that these findings indicate that the human visual system is attuned to balance perceptual omission of intrasaccadic motion with maintenance of sensitivity to moving stimuli.

This is an impressive piece of work. The manuscript is well written, the experiments have been carefully conducted and the results are largely convincing. While there are a few issues that need to be addressed, the study has the potential to be of significant interest to the field.

1. A feature of the results of Experiment 1 is that task performance tends towards chance when the stimulus speed approaches the saccade speed associated with a given movement amplitude (e.g. Figure 3b, relative speed = 1). While it is tempting to draw inferences from this regarding the perceptual omission of intrasaccadic motion, the fact that stimuli moved with constant speed makes it difficult to do this. Given that the link with perceptual omission during saccades is central to the authors' interpretation, it would be beneficial to see data from a supplementary experiment in which stimuli accelerate/decelerate to/from peak speed in a manner more comparable to saccade kinematics.
2. Throughout the experiments, the arc of the motion path is set such that the vertical amplitude is 15% of the horizontal amplitude. This seems like a somewhat arbitrary choice. Can we be confident that the relationship between task performance and relative speed would remain if a different arc had been chosen?
3. It is nice that results can be modelled (at least qualitatively) using a relatively simple spatiotemporal filtering account of early visual processing. While I don't think that it is necessary to fit the model to the data, I would like to see some consideration and discussion of which parameters are critical to reproducing the main experimental findings. In particular, how sensitive is model behavior to the scale of temporal and spatial blurring?
4. While the experimental task required participants to discriminate between upwards and downwards motion arcs, the model derives evidence from the difference in outputs for motion-present and motion-absent stimulus sequences. While I do not expect that it will substantially alter performance of the model, it would be more principled to match the implementation of model to the task at hand (i.e. derive evidence from the difference in outputs between upwards and downward motion arcs).
5. Results for the two correlational studies (Exp 3 & 4) could be more clearly presented. In Exp 3, the reported correlations reflect a mixture of covariation between thresholds and saccade speeds across amplitude, directions and individuals. Given that covariation between threshold and amplitude has already been established in Exp 1, it would be helpful to partition out this effect (e.g. calculate correlations for each amplitude separately). For both Exp 3 & 4, it would be useful to also see correlations for the two directions not shown (i.e. +/- 90 deg. relative to 'retinal' direction).

Reviewer #2

(Remarks to the Author)

The Rolfs et al study described in this manuscript examines the relationship between motion direction discrimination and saccades kinematics. In the main experiments described in the manuscript the authors replicate the retinal motion induced by saccades of different amplitudes in the absence of the actual eye movement. Subjects are asked to discriminate the vertical direction of motion of a stimulus. By systematically changing different variables, i.e., amplitude of the movement, duration and speed, the authors determine the visibility thresholds. These thresholds are then compared to the subjects' saccade kinematics. The authors find that there is a correlation between saccade speed and the visibility thresholds for speed, a similar correlation is reported for duration thresholds and saccade duration. It is concluded that the limits of motion direction discrimination follow the saccade kinematics. This evidence seems to align with the idea that the kinematics of action may constrain the visual system sensitivity to certain type of stimulation. Overall the work is interesting but it is not described clearly and it is not placed in the context of the literature, which makes it hard to assess its impact and significance.

1. The abstract needs to be rewritten to better reflect the content of the manuscript. The abstract is not very informative and it does not provide information on the actual findings and on the research methodology. In its current version the abstract may be misleading. It is centered on saccadic omissions, which entail retinal motion that is omitted from conscious perception, and on perceptual omission of the stimulus movement. This gives the reader the impression that the retinal motion generated during the saccade goes undetected. But the main findings of the study do not pertain to undetected retinal motion, rather the inability to judge the vertical component of motion of a stimulus mimicking a saccadic jump. Subjects could obviously still detect stimulus motion. The conclusive statement of the abstract, "Human vision and motor control are attuned to balance high visual sensitivity with perceptual omission of an action's sensory consequences" is also cryptic. What is meant by balancing high visual sensitivity with perceptual omissions?

2. Humans perform saccades 2-3 times per second, and every time a saccade is performed the whole visual world shifts at high speed on the retina, however, humans do not seem to perceive this motion, which would be perceptually unsettling. How is this possible? This is a century old problem that has fascinated scientists and it is still a very debated topic. One interpretation is that this motion is omitted because of retinal phenomena, i.e., masking, retinal speed too high to be perceived, the whole world moves and this triggers a motion suppression mechanism. Another interpretation is that the visuomotor system knows about the impending saccade through a corollary discharge signal and uses this signal to suppress motion perception during the saccade. The authors conclusion seems to support the idea that perceptual omissions during saccades are the result of retinal factors, yet, they do not rule out the possibility that extraretinal mechanisms are also at play. Although this debate is at the core of the question regarding the phenomenon of saccadic omissions/saccadic suppression, it is almost completely omitted from the manuscript with the exception a short paragraph in the discussion.

3. Related to point 2; there is virtually no mention of the extensive work by Burr and colleagues on saccadic suppression (eg., the mirror experiment in Diamond et al, 2000). Further, there is no mention of recent work from the Hafed group published in the same journal, showing that the saccadic suppression phenomenon has a retinal origin, and little mention of Zimmermann work on a similar topic also highlighting the contribution of visual factors to saccadic suppression. The introduction should mention this on-going debate and the relevant research and should describe how the present work fits in this context and how it is different.

4. Although, as stated by the authors, the results described here have "intriguing consequences for mainstream theories that rely on corollary discharge signals to explain perceptual experience" (lines 615-617), they do not rule out the possibility that a corollary discharge signal is also used in the context of saccadic suppression/omission (see also Hafed work in this regard). In fact, to rule this out one would need to stabilize the entire visual field during a saccade, removing completely the retinal signal associated with the saccade. If under these circumstances subjects cannot see the stimulus, this would show that the retinal signal is used to trigger saccadic suppression. However, this is a technically difficult experiment to conduct.

5. The authors' goal is to replicate the type of motion that the retina is exposed to during a saccade. Although the authors replicate the curved trajectory characterizing saccades, the simulated motion is extremely different from what humans experience during a saccade. First, a saccade is not simply a jump, retinal motion progressively accelerate, reaches a peak speed and decelerate. Further, saccadic overshoots/lens wobbling have also some impacts on retinal motion (see Taberner and Artal, 2014). This complex dynamic is not reproduced in the current study where the stimulus moves at a constant speed from the onset to the offset of the motion period, and leaves me wonder how this may influence the results reported here. The authors should at least explain their rationale for simulating saccade motion in this way and why they think that other components of saccadic motion can be safely ignored in this context.

6. The authors keep referring to visibility and visibility thresholds in their manuscript, which I find a bit misleading. It would be more accurate to define them as vertical motion direction discrimination thresholds as the stimulus itself was visible to the subject and the task was to discriminate the vertical component of motion (i.e., up vs down), I assume subjects could clearly detect and discriminate the horizontal component of motion.

7. The abstract states that perceptual omissions reflect the variability between individual observers, however, it is not clear where this is shown. What I would expect to see is evidence that variations in subjects' visibility thresholds correlate with

variations in their saccade peak speed, that is, subjects characterized by overall saccades with higher peak speeds have higher visibility thresholds, on the contrary subjects characterized by lower peak speeds, given the same saccade amplitude, have lower visibility thresholds. Is this that the authors are trying to convey here? Which figure shows this relationship? I think this would be compelling evidence in support of their model.

8. The part of the manuscript describing how visibility covaries with saccade kinematics seems to go back to the relationship between visibility thresholds and individual variations in saccade speed, but the link between these two things remains obscure to me. It clearly does not emerge from figure 4 where all saccade amplitudes are mixed. I think it would be useful to show a graph for a given saccade amplitude plotting on the x axis the average saccade speed for each subject across different directions, and on the y axis the corresponding average threshold speed. In figure 4 the correlation in the data seem to be primarily driven by saccade amplitude rather than by individual variability.

9. The authors delineate two possible outcomes. In one scenario the visibility thresholds do not depend on saccadic amplitude, in another scenario, these thresholds are proportional to the main sequence, i.e., they change with amplitude. Maybe I am missing something here, but I can't imagine a scenario in which speed thresholds do not depend on saccade amplitude. The larger the simulated saccade motion the larger is the vertical displacement of the stimulus (Fig. 2b). I would think that it is harder to discriminate the direction of motion for stimuli that move very fast and that it is easier to discriminate the direction of motion if the displacement of the stimulus is larger, assuming constant duration (as the case for saccades). Therefore, I would naturally expect that increasing the speed for a stimulus, which vertical displacement is small, will make the task harder, but at the same stimulus speed I may still be able to discriminate the direction of a larger displacement. I would be surprised if discrimination thresholds were independent from the movement amplitude. The authors should provide some rationale for the idea that speed threshold may be independent of the amplitude of the movement. On the other hand, I find the results reported in Fig. 5, showing that removing the stimulus presence before and after the motion leads to thresholds being independent from absolute amplitude, surprising. I wonder if the transient given by the sudden onset/offset of the stimulus is what primarily drives thresholds in the absence of the stimulus at the end and start point of the movement. I also think that these results should be expressed as a function of absolute, rather than relative speed.

Minor comments:

1. The tables in the main text are not particularly useful and make the text heavier, I would suggest moving them to supplemental material.

2. Figure 4; it is not clear how the correlation values are calculated. Were they based on the gray points in the graphs or on the across subjects averages? If correlations are based on the gray points each subject is represented multiple times (it seems 5 eccentricities and 4 directions). This would make the points non-independent and would require some kind of adjustment when calculating the correlation.

3. I find it a bit confusing that in the graphs relative speed is sometime defined as V/V_p , sometimes as V_p (which in the text is defined as peak velocity), and sometimes is referred as V_{rel} . Further, given that it is speed and not velocity, it would be better to call it S rather than V. Also, relative speed is not clearly defined in the text (see lines 82-86 vs. lines 129-131). Similarly, how was the absolute speed defined? It seems it is based on the peak speed for saccade of a given amplitude multiplied by a factor that is different for each amplitude. Why? This is also not clearly explained in the text.

4. In the graphs showing psychometric fits, are the dots with error bars average performance across subjects? Or do they represent performance when all subjects' trials are collapse together? If the former, reporting the number of trials in the graphs (e.g. 39745 trials) is not necessary. If the latter, it should be clearly specified in the text.

5. It is not clear what Figure 2d shows, is it meant to show that with different stimulus durations relative speed is modulated in proportion to V_p , essentially resulting in longer stimuli duration be characterized by lower speeds?

Reviewer #3

(Remarks to the Author)

The manuscript describes that properties of perpetual invisibility of a moving stimulus can be predicted by kinematics of that moving stimulus on the retina. This is relevant for understanding saccadic omission, i.e., perceptual omission of stimulus during eye movements such as saccades. Through human psychophysics, the authors demonstrate that a shared law links the limits of perceiving stimuli moving at high speed to the sensory consequences of rapid eye movements. They develop a parsimonious model to describe how this effect takes place.

While the experiments are quite thorough, I do have some concerns regarding the interpretability of the results, which requires further clarification. I detail them below:

1. The authors use the term "lawfully" to describe the relation between peak velocity and duration of saccades with movement amplitude. Can the authors clarify what they mean by lawful in their manuscript?

2. What is the basis of the null hypothesis used in this study, i.e. lines 122-125: "if visibility is simply predicted by absolute movement speed (v), then visibility thresholds should be independent of movement amplitude". But why should it be

independent of movement amplitude? If saccadic omission relates to a sensory consequence, then one would have to think in terms of activity of visual neurons. Larger amplitude movement of a stimulus means that the stimulus will traverse several receptive fields. The duration each receptive field is exposed to should be larger than a critical number in order for that neuron to be active. For a constant speed, large amplitudes should therefore require longer total durations in order to be visible. This is consistent with the results shown in Fig. 3c which shows that the total duration increases with increasing amplitude. This (the solid white lines in Fig. 3c) would be the expected behavior coming from a sensory consequence point of view. Maybe the authors can clarify the reasons/motivation for their null hypothesis.

3. If visibility thresholds depend on duration a stimulus is exposed, then perhaps looking at how visibility thresholds vary with the duration of exposure per unit amplitude might yield some interesting insights. For example, does the duration of exposure scale linearly with total amplitude or not?

4. Fig. 5a shows that in case there are no end-points, then visibility thresholds do not depend on the saccade main-sequence. So it appears that end-points are critical not just for rendering the motion invisible, but also in a way that it aligns with the main-sequence. But the way the paper mostly reads is that kinematics of retinal motion shape the visibility thresholds. Can the authors disambiguate this?

5. In the current study, constant stimulus velocity is used, meaning there is infinite acceleration/deceleration from the endpoints. It is likely that the acceleration/deceleration profile of end-points modulate the visibility thresholds given that they play a crucial role. Previous studies (eg Macknik and Livingstone 1998) have highlighted that the duration between stimulus and endpoint is relevant in rendering the stimulus invisible. Do the authors expect the velocity profile to modulate visual thresholds? Does this change any of their conclusions?

6. The finding that visibility covaries with saccade kinematics is very intriguing. The results here are presented as a support for the claim that visibility thresholds strongly depend on kinematics of retinal motion. Can the authors comment on why they think this is a stronger claim than the notion that saccade kinematics are tuned to render the motion invisible.

7. The model replicates the experimental findings. However in its current state, I find the model not very informative. For example, can the authors elaborate on where/or what processes in the brain might the different model components be analogous to? And how can the model be used to generate testable hypotheses for future studies? What do we actually learn from this model?

8. In the introduction and discussion, the authors mention that visibility thresholds relate to kinematics of retinal motion and this could be a consequence of eye movements. Do the authors mean that this relation is learnt over a human's lifetime or do they mean that this relation has evolved in species with eye movements? This could be a potential pointer for future studies to explore. For example, if it is learnt over a human's lifetime then presumably age of participants should matter. On the other hand if it evolved in species with eye movements then it would be interesting to see how visual thresholds relate to kinematics in species with less frequent eye movements

Version 1:

Reviewer comments:

Reviewer #1

(Remarks to the Author)

The authors should be commended for a rigorous and thoughtful response to previous comments. They have addressed my concerns and I am happy to recommend publication.

(Remarks on code availability)

Reviewer #2

(Remarks to the Author)

The authors did an excellent job in addressing the reviewers' concerns. The abstract has been revised and it is now more informative and links to previous work, before missing, have been highlighted. Most importantly, the authors conducted another experiment in which stimulus motion mimicked saccade-induced motion confirming the main conclusion.

I only have a couple of comments:

1. Given their importance in ruling out possible confoundings, the relatively large sample size, and the fact that Exp 2b replicates conditions of retinal motion that more closely resemble what experienced during actual saccades, and the fact that they directly refer to saccadic omissions, I think the supplementary results of Figure S3 (Exp 2b) should be included in the main text, and it should be explained why the results for relative movement speed do not perfectly match with the predictions. Ideally, Exp 2b should be the main experiment. It would also be important to perform analyses similar to those of Fig 4 for experiment 2b.

2. The authors mention that the retinal motion introduced by post saccadic lens wobbling is negligible, however Tabernero and Artal analyses, estimating the actual retinal motion on the cone mosaic resulting from lens wobbling, suggest otherwise (max retinal displacement of 0.3 deg for a 9 deg saccade). This wobbling is particularly consequential for the stimulus the saccade lands on, the stimulus at the very center of gaze. Here, cone receptors are most closely spaced, and even a small amount of motion can have important perceptual consequences. I am not asking the authors to replicate this motion with an additional control experiment, but they should at least acknowledge and discuss the potential relevance of this retinal motion component that is completely ignored in their model and in their simulated motion trajectories. How can this additional component of retinal motion influence perception and possibly the pattern of results shown here?

(Remarks on code availability)

Reviewer #3

(Remarks to the Author)

I thank the authors for their detailed replies and explanations. The manuscript definitely has improved.

In reference to my comment #4 on the previous submission, that end-points appear to be critical not just for rendering the motion invisible, but also in a way that it aligns with the main-sequence. I understand that in the current experimental setting, it is not possible to explicitly disambiguate the extent to which end-points play a role in rendering motion invisible in a way dependent on the main sequence. The authors have now addressed this briefly in the relevant results section. It will be helpful if the authors make this limitation and their underlying assumptions and its justification more explicit in the discussion section.

(Remarks on code availability)

Version 2:

Reviewer comments:

Reviewer #2

(Remarks to the Author)

The authors addressed all the remaining comments.

(Remarks on code availability)

Reviewer 1:

This manuscript describes a set of experiments examining the relationship between the limits of visibility for fast moving stimuli and regularities in the retinal motion produced by saccadic eye movements. Upper-speed thresholds for detecting curvature in the path of translating Gabor patches are shown to scale with the spatial extent of motion, in a manner that is predictable consistent with the main sequence for saccades. Speed thresholds are also shown to systematically covary with saccade kinematics across individuals and manipulations of direction/amplitude. A model of early visual processing incorporating spatial and temporal blur filters is developed that reproduces the scaling of speed thresholds with spatial extent of motion, along with the finding that this relationship only holds when the stimulus remains visible at the start & endpoints of the trajectory. The authors argue that these findings indicate that the human visual system is attuned to balance perceptual omission of intrasaccadic motion with maintenance of sensitivity to moving stimuli.

This is an impressive piece of work. The manuscript is well written, the experiments have been carefully conducted and the results are largely convincing. While there are a few issues that need to be addressed, the study has the potential to be of significant interest to the field.

We thank the reviewer for appreciating the extent and potential reach of this work. We believe that our revisions have made the manuscript even stronger and thank the reviewer for their helpful feedback.

1. A feature of the results of Experiment 1 is that task performance tends towards chance when the stimulus speed approaches the saccade speed associated with a given movement amplitude (e.g. Figure 3b, relative speed = 1). While it is tempting to draw inferences from this regarding the perceptual omission of intrasaccadic motion, the fact that stimuli moved with constant speed makes it difficult to do this. Given that the link with perceptual omission during saccades is central to the authors' interpretation, it would be beneficial to see data from a supplementary experiment in which stimuli accelerate/decelerate to/from peak speed in a manner more comparable to saccade kinematics.

Indeed, performance in the curvature discrimination task approaches chance level at around saccadic speed. We caution, however, to overinterpret this result. As we pointed out in the Discussion of our initial submission, we would expect this threshold to vary as a function of various factors (e.g., stimulus contrast or spatial frequency), but importantly, in a way that retains their proportionality to the main sequence.

We appreciate the reviewer's interest in seeing an additional experiment in which stimuli follow a velocity profile that is comparable to saccade kinematics, a point that has also been put forward by Reviewer #3, and alluded to by Reviewer #2. We have run this experiment as a detection task, in which participants ($N = 10$, 4200 trials each) reported whether the stimulus moved continuously (motion present) or instead, jumped (motion absent) from its initial to its final location. We have added this new experiment as Experiment 2b to the **Supplementary material**, and refer to it in several places in the main text. We provide a brief summary along with the new **Supplementary Figure S3** here:

The trial procedure (**Figure S3a**) was largely modeled after Experiment 2a (Experiment 2 in the previous version of the manuscript), using 5 movement amplitudes and 7 relative movement speeds (ranging from 0.25 to 2 times saccadic peak velocity): The stimulus either moved continuously, and on a straight horizontal path, from its initial to its final location (motion present, 50% of all trials) or jumped to its new location (motion absent, 50% of all trials). On motion present trials, stimulus speed mimicked the velocity profile of a saccade (**Figure S3b-c**; see **Methods** for details). On stimulus-absent trials, to prevent observers from using sudden onsets and offsets to establish the absence of motion, we rapidly ramped down the stimulus contrast to zero, when it jumped from to its final location, before rapidly ramping up to full contrast again. As in Experiment 2a, participants reported whether the stimulus moved continuously (motion present) or instead, jumped (motion absent) to its new location.

Given the acceleration and deceleration phases of a saccadic velocity profile scale with movement amplitude, we predicted thresholds for *relative* movement speed to depend moderately on movement distance (link to pre-registration in Methods). This prediction is also consistent with imperfect saccadic omission during saccades, which results from slow phases of the saccadic velocity profile (Schweitzer et al., 2023). If it were met, it would add further evidence that the key results of our study are relevant to understanding saccadic omission.

Figure S3 | Motion detection thresholds for saccadic velocity profiles (Exp. 2b). **a** Trial procedure. Observers judged whether the stimulus jumped (Motion absent) or moved continuously on a straight horizontal path (Motion present) from its initial to its final location. **b** Absolute speed for each movement amplitude when relative movement speed was 1. **c** Relative movement speeds for 4 dva amplitude. **d-e** Performance (left column) and visibility thresholds (right column) resulting from best-fitting Naka-Rushton functions as a function of absolute and relative movement speed, respectively, plotted for each amplitude.

Indeed, performance showed a clear dependence on movement distance and speed, compatible with the main sequence. Expressed as a function of *absolute* stimulus speed,

performance systematically increased with movement amplitude, shifting psychometric functions systematically to the right (**Figure S3d**, left panel). Visibility thresholds (right panel) again closely followed our prediction (solid white lines), with slightly steeper slopes and remarkable consistency across observers (gray lines). Expressed as a function of *relative* movement speed, the distance between psychometric functions decreased markedly, and thresholds settled around 60 to 80% of saccadic peak velocity (**Figure S3e**, right panel). As predicted, there remained a small but significant increase with movement amplitude, which was clearly inconsistent with the prediction of a constant velocity threshold (dashed white lines). These results confirm that visibility thresholds systematically depend on movement amplitude in a way well-described by the main-sequence relation of saccadic eye movements.

2. Throughout the experiments, the arc of the motion path is set such that the vertical amplitude is 15% of the horizontal amplitude. This seems like a somewhat arbitrary choice. Can we be confident that the relationship between task performance and relative speed would remain if a different arc had been chosen?

We are highly confident that the choice of the arc is largely irrelevant to the results. Experiment 2 in the initial submission (now Experiment 2a) used a detection task, in which observers had to distinguish the presence from the absence of continuous motion. In this experiment, continuous motion did not follow an arc, but a perfectly horizontal path. Yet, the results were consistent with the results in the curvature discrimination task: We observed the same proportionality of visibility thresholds to the main sequence. Admittedly, Experiment 2 was previously only briefly mentioned in the main text. As pointed out above, we have now also added a second experiment (**Exp. 2b**) that uses a detection task and no curvature whatsoever. For space constraints, we would like to keep their full description in the Supplementary material, but we now dedicate a full paragraph to them in the main text, ensuring that this generalization of our results does not go unnoticed in the revised manuscript:

We replicated these results in a detection task in which, instead of discriminating the curvature of the stimulus' motion path, observers reported the presence or absence of continuous motion on a straight horizontal path. In this version of the task, we moved stimuli either with constant speed as in Experiment 1 (**Supplementary Material**, Exp. 2a) or with time-varying speed, mimicking the velocity profile of saccades (Exp. 2b). These supplementary experiments confirmed that visibility thresholds systematically depend on movement amplitude in a way well-described by the main-sequence relation of saccadic eye movements. Moreover, they show that this result is not merely an artifact of the curvature task (e.g., due to the vertical motion component). Finally, they confirm that our results are relevant to saccadic omission^{32,34}. While saccades impose time-varying velocity profiles onto the retinal surface (as in Exp. 2b), key kinematic parameters of saccades—amplitude, velocity, and duration—suffice to predict visibility of stimulus motion.

3. It is nice that results can be modelled (at least qualitatively) using a relatively simple spatiotemporal filtering account of early visual processing. While I don't think that it is necessary to fit the model to the data, I would like to see some consideration and discussion of which parameters are critical to reproducing the main experimental findings. In particular, how sensitive is model behavior to the scale of temporal and spatial blurring?

To provide an answer to this important remark, we performed a large-scale grid search on the crucial model parameters, that is, σ_{RF} (the standard deviation of the spatial Gaussian kernel), θ_{RF} (the scale of the temporal Gamma kernel), and σ_E (the standard deviation of the Gaussian sampling to simulate binary responses from sensory evidence), and documented its results in the section ‘Grid-search analyses’ in the Supplementary Material. To go beyond a mere qualitative approach, we determined the models’ goodness of fit by comparing experimentally determined velocity thresholds with thresholds predicted by the model, obtained by fitting simulated model responses with the same psychometric function as the experimental data.

As shown in **Figure S6** (copied below), we found that the varied model parameters had a profound impact on model predictions and that the parameters we chose for the primary analysis (all motivated by published neurophysiological data) were close to the optimum. First, we found that the critical parameter to achieve the reported model behavior was to use a reasonably high θ_{RF} (that is, in our analysis around 18), which is, in fact, a value that can describe the temporal dynamics of simple cells in primary visual cortex, when fitting post-stimulus response histograms with the Gamma function (Schweitzer et al., 2023). Second, with a suitable θ_{RF} and if the Gaussian aperture of the stimulus was modeled, then a spatial filtering stage in the model was not even necessary, as even σ_{RF} as low as 0.05 degrees of visual angle produced good model predictions (**Figure S6c**). Third, to quantitatively approach empirical velocity thresholds, the model readout stage plays an important role. Specifically, sharper temporal response functions (lower θ_{RF}) result in higher overall evidence for motion. In such cases, increased readout noise σ_E is needed to prevent unrealistically high velocity thresholds (**Figure S6d**). Even though we selected σ_E parameters that would be optimal for each θ_{RF} - σ_{RF} combination, only a specific parameter subspace produced model predictions that qualitatively matched experimental results (**Figure S6e**). These results were confirmed by a final analysis that investigated how well the signature result of Experiment 5—the similarity of velocity thresholds across amplitudes at longer pre- and post-movement stimulus duration—could be reproduced: Again, virtually the same parameter space caused predicted thresholds to converge (**Figure S6f**).

Figure S6 | Grid-search analysis to determine model parameter space. **a** Effects of changing the Gamma scale θ_{RF} and the Gaussian SD σ_{RF} on temporal (upper panel) and spatial (lower panel) response functions, respectively. **b** Results of the grid search when using a fixed SD $\sigma_E=4$ for the Gaussian sampler applied to convert sensory evidence E to model responses X (see Methods). Colors represent MSE between empirically measured velocity thresholds and velocity thresholds predicted by the model. **c** Results of the grid search in terms of MSE when an optimal σ_E is selected for each grid point. The same color scale as in panel b is used. **d** Resulting optimal σ_E for each grid point displayed in panel c. **e** Three examples of velocity thresholds predicted by the model as a function of amplitude and pre- and post-movement stimulus duration. Shaded areas represent ± 2 SEM around the average thresholds found in Experiment 5. **f** Mean absolute deviation of estimated thresholds for 50 ms and 200 ms of pre- and post-movement stimulus duration. Low values indicate little or no difference between velocity thresholds across movement amplitudes, that is, the signature finding in Experiment 5.

4. While the experimental task required participants to discriminate between upwards and downwards motion arcs, the model derives evidence from the difference in outputs for motion-present and motion-absent stimulus sequences. While I do not expect that it will substantially alter performance of the model, it would be more principled to match the implementation of model to the task at hand (i.e. derive evidence from the difference in outputs between upwards and downward motion arcs).

The reviewer is right that the task given to the model does not affect the predictions of the model. Indeed, the model documented in the manuscript is essentially comparing the evidence for stimulus above the midline (that shows continuous motion between endpoints) to the evidence for a stimulus below the midline (that shows only endpoints). The model's binary response is then based on that difference. We have clarified that this is the case in the revised version of the manuscript.

Note that we report the Gaussian sampling procedure in the manuscript because it is closest to the 2AFC task applied in the experimental task and because it only requires one parameter (i.e., σ_E). Nevertheless, we also implemented a version of the model in which the model compares

the evidence for motion against a threshold. This so-called binomial sampler first creates a normal distribution around the mean evidence (with standard deviation σ_E) and then for every value applies a threshold k – an additional parameter representing an absolute sensory threshold – above which responses are sampled binomially with $p=1$ -lapse rate and below which responses are sampled binomially with $p=0.5$. With the right settings this sampling procedure produces virtually the same results, which can be inspected in the markdown document made publicly available at

https://github.com/richardschweitzer/ModelingVisibilityOfSaccadelikeMotion/blob/main/SLM_model_ver1.md.

5. Results for the two correlational studies (Exp 3 & 4) could be more clearly presented. In Exp 3, the reported correlations reflect a mixture of covariation between thresholds and saccade speeds across amplitude, directions and individuals. Given that covariation between threshold and amplitude has already been established in Exp 1, it would be helpful to partition out this effect (e.g. calculate correlations for each amplitude separately). For both Exp 3 & 4, it would be useful to also see correlations for the two directions not shown (i.e. +/- 90 deg. relative to 'retinal' direction).

We understand the concern that the correlation between visibility thresholds and saccadic velocity is mostly driven by movement amplitude, which is a strong covariate of both these measures. For completeness, we would first like to clarify that we had already addressed this point in our initial submission in three separate ways: First, we compared these correlations for two directions (retinal vs spatial), both of which have the same strong correlation with movement amplitude. Yet, thresholds in the retinal direction consistently showed higher correlations to saccade velocity than thresholds in the spatial direction (**Figure 4**, side panels). Second, we performed stepwise regressions that showed that the contribution of retinal speed is still significant when the effects of amplitude and direction were taken into account (**Supplementary Material**). This analysis statistically partitions out the effect of movement amplitude, as suggested by the Reviewer. Third, we conducted Experiment 4, which focused on a single amplitude in a larger group of participants and showed correlations of saccade velocity with thresholds in the retinal (but not the spatial) direction (**Figure 4**, right panels), effectively ruling out an effect of amplitude.

We have now conducted three more analyses, each of which made us more confident in our conclusions. First, we show that regression models using retinal direction as a predictor of visibility thresholds consistently outperformed models using spatial direction. This was true for both Experiments 3 and 4, and was consistent across analyses of speed and duration. Second, as requested by the reviewer, we calculated the same stepwise regression models using the two directions not shown in the manuscript (± 90 off the retinal direction). Again, we see that the retinal model accounts for the visibility threshold data consistently better than either the +90 or the -90 deg predictors. We have added the results of these to the **Supplementary Material** (*Experiments 3 and 4: Stepwise regressions*).

Third, we have followed the reviewer's suggestion to partition out the effect of amplitude from the correlations in Experiment 3. For each movement amplitude, we have removed the mean

across all subjects from each of our dependent variables (visibility thresholds for absolute movement speed and duration, saccade peak velocity and saccade duration). We then determined correlations for the residuals of thresholds and movement parameters. After controlling for the impact of movement amplitude in this way, we found that the peak velocity of saccades still correlated significantly with visibility thresholds for absolute movement speed in the opposite (retinal) direction (Spearman's $\rho = 0.40$, $p < 0.001$), but not in the spatial direction ($\rho = -0.07$, $p = 0.466$). Similarly, the duration of saccades still correlated significantly with visibility thresholds for absolute movement duration in the opposite (retinal) direction ($\rho = 0.41$, $p < 0.001$), but not in the spatial direction ($\rho = -0.05$, $p = 0.594$). We have added these additional analyses to the corresponding part of the **Results** section (*Visibility covaries with saccade kinematics*).

Reviewer 2:

The Rolfs et al study described in this manuscript examines the relationship between motion direction discrimination and saccades kinematics. In the main experiments described in the manuscript the authors replicate the retinal motion induced by saccades of different amplitudes in the absence of the actual eye movement. Subjects are asked to discriminate the vertical direction of motion of a stimulus. By systematically changing different variables, i.e., amplitude of the movement, duration and speed, the authors determine the visibility thresholds. These thresholds are then compared to the subjects' saccade kinematics. The authors find that there is a correlation between saccade speed and the visibility thresholds for speed, a similar correlation is reported for duration thresholds and saccade duration. It is concluded that the limits of motion direction discrimination follow the saccade kinematics. This evidence seems to align with the idea that the kinematics of action may constrain the visual system sensitivity to certain type of stimulation. Overall the work is interesting but it is not described clearly and it is not placed in the context of the literature, which makes it hard to assess its impact and significance.

We thank the reviewer for their interest in this work and for pointing out aspects of the manuscript that could elevate its clarity and more strongly embed it in the existing literature.

1. The abstract needs to be rewritten to better reflect the content of the manuscript. The abstract is not very informative and it does not provide information on the actual findings and on the research methodology. In its current version the abstract may be misleading. It is centered on saccadic omissions, which entail retinal motion that is omitted from conscious perception, and on perceptual omission of the stimulus movement. This gives the reader the impression that the retinal motion generated during the saccade goes undetected. But the main findings of the study do not pertain to undetected retinal motion, rather the inability to judge the vertical component of motion of a stimulus mimicking a saccadic jump. Subjects could obviously still detect stimulus motion. The conclusive statement of the abstract, "Human vision and motor control are attuned to balance high visual sensitivity with perceptual omission of an action's sensory consequences" is also cryptic. What is meant by balancing high visual sensitivity with perceptual omissions?

We thank the reviewer for this helpful feedback. We have now taken more space to explain the results and conclusions clearly in the abstract.

We would like to avoid a possible misunderstanding though: The reviewer points out that our main findings "do not pertain to undetected retinal motion, rather the inability to judge the vertical component of motion of a stimulus mimicking a saccadic jump" and continues "Subjects could obviously still detect stimulus motion." To avoid confusion, we need to distinguish between detecting continuous motion from detecting a change in the stimulus' position (i.e., a simple displacement). As the reviewer points out, there is no doubt that observers could see the stimulus change position (e.g., from the left to the right of fixation) even at the highest retinal speeds. Observers could not, however, detect the continuous motion itself at any high velocity, neither the vertical nor the (more relevant) horizontal component. This result is most apparent in the previously reported in Experiment 2 (now Experiment 2a) in the **Supplementary Material**,

which was admittedly only briefly mentioned in the main text. In this version of the experiment, the stimulus had no vertical component. Instead, observers had to judge the presence (vs. absence) of continuous motion on a straight horizontal path. In the revision, we have also added a second detection experiment (Experiment 2b) using saccadic velocity profiles rather than constant speeds. This experiment shows that observers are unable to distinguish continuous motion at high speeds from a simple displacement of the stimulus. More importantly, we replicate the key finding that visibility thresholds are predicted by the main sequence. We now dedicated a full paragraph to these experiments in the main text, ensuring that these aspects of our results are readily seen in the revised manuscript:

We replicated these results in a detection task in which, instead of discriminating the curvature of the stimulus' motion path, observers reported the presence or absence of continuous motion on a straight horizontal path. In this version of the task, we moved stimuli either with constant speed as in Experiment 1 (**Supplementary Material**, Exp. 2a) or with time-varying speed, mimicking the velocity profile of saccades (Exp. 2b). These supplementary experiments confirmed that visibility thresholds systematically depend on movement amplitude in a way well-described by the main-sequence relation of saccadic eye movements. Moreover, they show that this result is not merely an artifact of the curvature task (e.g., due to the vertical motion component). Finally, they confirm that our results are relevant to saccadic omission^{32,34}. While saccades impose time-varying velocity profiles onto the retinal surface (as in Exp. 2b), key kinematic parameters of saccades—amplitude, velocity, and duration—suffice to predict visibility of stimulus motion.

2. Humans perform saccades 2-3 times per second, and every time a saccade is performed the whole visual world shifts at high speed on the retina, however, humans do not seem to perceive this motion, which would be perceptually unsettling. How is this possible? This is a century old problem that has fascinated scientists and it is still a very debated topic. One interpretation is that this motion is omitted because of retinal phenomena, i.e., masking, retinal speed too high to be perceived, the whole world moves and this triggers a motion suppression mechanism. Another interpretation is that the visuomotor system knows about the impending saccade through a corollary discharge signal and uses this signal to suppress motion perception during the saccade. The authors conclusion seems to support the idea that perceptual omissions during saccades are the result of retinal factors, yet, they do not rule out the possibility that extraretinal mechanisms are also at play. Although this debate is at the core of the question regarding the phenomenon of saccadic omissions/saccadic suppression, it is almost completely omitted from the manuscript with the exception a short paragraph in the discussion.

Our study was motivated by the idea that saccades have stereotyped kinematics that impose reliable sensory consequences on the visual system, which—given their high rate in natural vision—may impose constraints on visual processing (cf. Rolfs & Schweitzer, 2022). With this idea in mind, we predicted that perceptual omission of motion could systematically relate to the amplitude-velocity-duration relation of saccadic eye movements. We agree with the reviewer that this idea, as well as our findings, directly speak to the issue of saccadic omission and suppression, and we now discuss existing theories more prominently in the manuscript. We have added the following paragraph in the Introduction:

A broad range of accounts has been put forward for this reliable absence of perceiving the sensory consequences of saccades^{10,35,36}, invoking mechanical^{37,38}, retinal^{24,39,40}, and extraretinal mechanisms^{31,41,42}. While there is consistent evidence for the saccade-locked reduction of visual sensitivity (especially to low spatial frequencies⁴²⁻⁴⁴ which should remain well resolvable at high saccadic speeds⁴⁵) perception of motion during saccades is well possible within a resolvable temporal-frequency range^{21,23,25}. These findings can be reconciled by a constant visibility threshold at some temporal frequency^{46,47} (or speed, for a given spatial frequency^{24,45,48}) beyond which a stimulus becomes invisible. Saccades often have small movement amplitudes that induce lower retinal velocities, thereby allowing for a broader range of resolvable spatial frequencies¹⁵, yet the visual consequences of these eye movements are routinely omitted from conscious perception. We thus investigated if the limits of visibility of stimuli at high speed are predicted by saccade-related metrics, specifically those described by the main sequence. Such a relationship would suggest that the kinematics of the retinal image, lawfully induced by saccades, could shape the profile by which moving stimuli are omitted from perception.

We also expanded the relevant paragraph in the Discussion:

Experimental evidence that predictions based on sensorimotor contingencies play a role in saccadic omission has recently been presented by demonstrating that the strength of intra-saccadic motion percepts could be downregulated after habituation⁹⁰. Because our proposed mechanism does not strictly require the idea of a corollary-discharge-based prediction, it is a parsimonious explanation of the current results that were obtained during fixation in the absence of saccades, as well as saccadic omission in natural vision. It does not, however, constitute an explanation for the reduction of contrast sensitivity for briefly flashed gratings, which are widely used to characterize contrast sensitivity during saccades⁴²⁻⁴⁴. Recent neurophysiological evidence suggests, however, that these effects may be caused by visual mechanisms as well: Contrast sensitivity reduction can be observed in retinal ganglion and bipolar cells, thus clearly in the absence of influences from corollary discharge³⁹. While this does not exclude an impact of extra-retinal signals in saccadic omission, their role might be different than previously assumed. For instance, they could enhance contrast sensitivity upon saccade landing³⁹, or they might be part of a sensorimotor contingency that is used to down-regulate the visual consequences of eye movements³¹.

However, compared to a simple reduction in visual sensitivity (whether of retinal or extra-retinal origin), a mechanism based on the kinematics of saccades seems appealing: It would allow for the perceptual omission of motion during saccades while maintaining maximal sensitivity to high-speed motion during fixation (with a boundary defined by saccade kinematics). This retained sensitivity comes with at least two benefits: First, the visual system would miss little (potentially relevant) motion in the world while largely ignoring motion caused by one's own eye movements. And, second, residual sensitivity to the slower phases of saccade-induced motion would bleed through³⁴, which may help tracking objects' swiftly changing positions on the retina when the eyes move³⁰.

3. Related to point 2; there is virtually no mention of the extensive work by Burr and colleagues on saccadic suppression (eg., the mirror experiment in Diamond et al, 2000). Further, there is no mention of recent work from the Hafed group published in the same journal, showing that the saccadic suppression phenomenon has a retinal origin, and little

mention of Zimmermann work on a similar topic also highlighting the contribution of visual factors to saccadic suppression. The introduction should mention this on-going debate and the relevant research and should describe how the present work fits in this context and how it is different.

We have included these references in the introduction of the revised manuscript and describe how the present work adds to this ongoing debate (see our reply to the previous point).

4. Although, as stated by the authors, the results described here have “intriguing consequences for mainstream theories that rely on corollary discharge signals to explain perceptual experience” (lines 615-617), they do not rule out the possibility that a corollary discharge signal is also used in the context of saccadic suppression/omission (see also Hafed work in this regard). In fact, to rule this out one would need to stabilize the entire visual field during a saccade, removing completely the retinal signal associated with the saccade. If under these circumstances subjects cannot see the stimulus, this would show that the retinal signal is used to trigger saccadic suppression. However, this is a technically difficult experiment to conduct.

We now discuss the relevance of corollary discharge signals more extensively in the revised Discussion section of our manuscript (see above). In doing so, we also highlight the challenges of isolating the impact of any single influence (retinal motion, corollary discharge, and others) on perception:

Delineating the unique contributions of various signals to saccadic omission constitutes an exciting line of research that, of course, comes with severe methodological challenges (e.g., to stabilize the whole-field visual input during saccades; to faithfully replay the visual consequences of saccades during fixation; to equate the deployment of visual attention across saccade and fixation conditions; etc.)¹⁰.

Note that we believe there might have been a misplaced negation / typo in the reviewers' comment. The reviewer points out that “If under these circumstances subjects cannot see the stimulus, this would show that the retinal signal is used to trigger saccadic suppression.” Our impression is that this is either missing a “not” or has one too many. The condition that the reviewer describes is one in which the saccade has no retinal consequences whatsoever. Thus, the prediction should either be: “If under these circumstances subjects cannot see the stimulus, this would show that the retinal signal is NOT used to trigger saccadic suppression”, or “If under these circumstances subjects CAN see the stimulus, this would show that the retinal signal is used to trigger saccadic suppression”. We have revised the manuscript with this assumption in mind. We apologize and hope for clarification if we misunderstood this point.

5. The authors' goal is to replicate the type of motion that the retina is exposed to during a saccade. Although the authors replicate the curved trajectory characterizing saccades, the simulated motion is extremely different from what humans experience during a saccade. First, a saccade is not simply a jump, retinal motion progressively accelerates, reaches a peak speed and decelerates. Further, saccadic overshoots/lens wobbling have also some impacts on retinal motion (see Taberner and Artal, 2014). This complex dynamic is not reproduced in the

current study where the stimulus moves at a constant speed from the onset to the offset of the motion period, and leaves me wonder how this may influence the results reported here. The authors should at least explain their rational for simulating saccade motion in this way and why they think that other components of saccadic motion can be safely ignored in this context.

In the main set of experiments (Experiments 1, 2a, 3, 4 and 5), we have focused on replicating the main sequence relation (amplitude-velocity-duration) in our stimulus. Indeed, the visual consequences of saccades are more complex. As pointed out in response to this reviewer's first point, we have now added a new experiment in which the stimulus' speed follows the velocity profile of saccadic eye movements. The results of this experiment are summarized in the main text and fully presented in the **Supplementary Material** (*Experiment 2b: Visibility thresholds for detection of high-speed motion with saccadic velocity profiles*). A short summary as well as the relevant **Supplementary Figure S3** can also be found above in response to point 1 of Reviewer 1.

Note that we did not include post-saccadic oscillations in these simulated movements. Based on psychophysical data, Deubel et al. (1995) estimated the perceptual shift of the stimulus due to the lens wobble to be about 0.062 dva per degree of overshoot and measured a maximum overshoot amplitude of 4 deg for 12-deg saccades, which would correspond to a maximum perceptual shift of 0.248 dva for the largest movement amplitude on our display. We point this out in the methods section of the revised manuscript:

Note that the simulated movements did not include post-saccadic oscillations⁶², which have reliable yet comparably small perceptual consequences⁹³.

6. The authors keep referring to visibility and visibility thresholds in their manuscript, which I find a bit misleading. It would be more accurate to define them as vertical motion direction discrimination thresholds as the stimulus itself was visible to the subject and the task was to discriminate the vertical component of motion (i.e., up vs down), I assume subjects could clearly detect and discriminate the horizontal component of motion.

We thank the reviewer for pointing out this potential source of confusion. As explained in response to this reviewer's first point, observers could indeed clearly see that the stimulus changed position from before to after the movement. They could not, however, (neither subjectively or objectively) see the continuous horizontal motion of the stimulus as it changed position. Supplementary Experiments 2a and 2b eliminated vertical motion altogether, confirming that visibility was not based on discriminating the vertical component of motion. In revising the manuscript, we have clarified this in any instance in the manuscript where we thought it could be ambiguous.

7. The abstract states that perceptual omissions reflect the variability between individual observers, however, it is not clear where this is shown. What I would expect to see is evidence that variations in subjects' visibility thresholds correlate with variations in their saccade peak speed, that is, subjects characterized by overall saccades with higher peak speeds have higher visibility thresholds, on the contrary subjects characterized by lower peak speeds, given

the same saccade amplitude, have lower visibility thresholds. Is this that the authors are trying to convey here? Which figure shows this relationship? I think this would be compelling evidence in support of their model.

The reviewer hopes to see evidence that variations in visibility thresholds correlate with variations in peak velocity of saccades across individuals. Indeed, we go one step further by showing that the correlation is specific to the direction of retinal motion that a saccade entails rather than to saccadic peak velocity per sé. This correlation is shown in **Figure 4** and backed up by detailed statistical analyses (e.g., stepwise hierarchical regression models controlling for the impact of saccade amplitude and direction) in the **Supplementary Material**. We have now added additional evidence for this specificity (see next point). We agree that this correlation is compelling support for our hypothesis.

8. The part of the manuscript describing how visibility covaries with saccade kinematics seems to go back to the relationship between visibility thresholds and individual variations in saccade speed, but the link between these two things remains obscure to me. It clearly does not emerge from figure 4 where all saccade amplitudes are mixed. I think it would be useful to show a graph for a given saccade amplitude plotting on the x axis the average saccade speed for each subject across different directions, and on the y axis the corresponding average threshold speed. In figure 4 the correlation in the data seem to be primarily driven by saccade amplitude rather than by individual variability.

We were also wondering if the correlation between visibility thresholds and individual variations in saccade kinematics is driven by movement amplitude alone (a similar point was also raised by Reviewer 1, see their point 5). Four striking pieces of evidence show that movement amplitude does not explain the full extent of these correlations:

(1) We compared these correlations for two directions (retinal vs spatial), both of which have the same strong correlation with movement amplitude. Yet, thresholds in the retinal direction consistently showed higher correlations to saccade velocity than thresholds in the spatial direction (**Figure 4**, side panels). We consider this finding stronger evidence for our hypothesis than a mere correlation between individual visibility thresholds and saccade speed (as requested by the reviewer), because it isolates the effect of a saccade's retinal consequence on visibility threshold from a general performance factor. The main panels in Figure 4 show this relation between the peak velocity of *retinal movement* during saccades (on the abscissa/x-axis) and visibility thresholds (on the ordinate/y-axis).

(2) Importantly, controlling for the impact of movement amplitude by partitioning out the effect of movement amplitude, the peak velocity of saccades still correlated significantly with visibility thresholds for absolute movement speed in the opposite (retinal) direction (Spearman's $\rho = 0.40$, $p < 0.001$), but not in the spatial direction ($\rho = -0.07$, $p = 0.466$). We have added these new analyses to the corresponding part of the Results section (Visibility covaries with saccade kinematics). For your convenience, here are the corresponding parts of the results section for speed and duration, respectively:

Note that significant correlations in Experiment 3 were not solely driven by the main sequence itself: Even after removing the main effect of saccade amplitude by removing the across-subjects mean of each of the dependent variables, we still found that saccadic peak velocity correlated

significantly with visibility thresholds for absolute movement speed in the opposite (retinal) direction ($\rho = 0.40$, $p < 0.001$), whereas the correlations for the spatial direction vanished ($\rho = -0.07$, $p = 0.466$).

[...]

Again, for the retinal direction in Experiment 3, these correlations remained significant even after removing the effect of saccade amplitude ($\rho = 0.41$, $p < 0.001$), while they did not for the spatial direction ($\rho = -0.05$, $p = 0.594$).

(3) Experiment 4 focused on a single amplitude, yet considerable correlations of saccade velocity (and duration) with thresholds in the retinal (but not the spatial) direction remained (**Figure 4**, right panels), ruling out an effect of amplitude by experimental design.

(4) We performed stepwise regressions that showed that the contribution of retinal speed is still significant when the effects of amplitude (and direction) were taken into account (**Supplementary Material**).

9. The authors delineate two possible outcomes. In one scenario the visibility thresholds do not depend on saccadic amplitude, in another scenario, these thresholds are proportional to the main sequence, i.e., they change with amplitude. Maybe I am missing something here, but I can't imagine a scenario in which speed thresholds do not depend on saccade amplitude. The larger the simulated saccade motion the larger is the vertical displacement of the stimulus (Fig. 2b). I would think that it is harder to discriminate the direction of motion for stimuli that move very fast and that it is easier to discriminate the direction of motion if the displacement of the stimulus is larger, assuming constant duration (as the case for saccades). Therefore, I would naturally expect that increasing the speed for a stimulus, which vertical displacement is small, will make the task harder, but at the same stimulus speed I may still be able to discriminate the direction of a larger displacement. I would be surprised if discrimination thresholds were independent from the movement amplitude. The authors should provide some rationale for the idea that speed threshold may be independent of the amplitude of the movement. On the other hand, I find the results reported in Fig. 5, showing that removing the stimulus presence before and after the motion leads to thresholds being independent from absolute amplitude, surprising. I wonder if the transient given by the sudden onset/offset of the stimulus is what primarily drives thresholds in the absence of the stimulus at the end and start point of the movement. I also think that these results should be expressed as a function of absolute, rather than relative speed.

First, to avoid a potential misunderstanding, we now more clearly point out that our results are not merely driven by the extent of vertical motion in the curvature discrimination task. As pointed out above, the result that velocity thresholds depend on motion amplitude was found in a virtually identical manner in a detection task (Experiment 2a) where no vertical component was present (see response to Point 2 of Reviewer 1). It is thus highly unlikely that larger vertical motion components counteract the effect of higher speeds at larger (horizontal) motion amplitudes.

Second, we have now clarified the rationale of the constant threshold predictions. That is, it is more parsimonious to assume that thresholds solely depend on the temporal sensitivity profile

of the visual system, which has been extensively studied. Unstabilized full contrast thresholds of gratings moving behind a static aperture (which largely follow the same temporal sensitivity as flickering gratings; Kelly, 1979), were found to be around 50 Hz across a range of spatial frequencies (Kelly, 1966). This finding extended to moving objects: Burr & Ross (1982) found sinusoidal bars of 1 dva width became invisible at velocities between 50 and 100 dva/s. We reasoned, therefore, that in the absence of other moderating processes detection should be purely dependent on the system's temporal tuning and the temporal-frequency spectrum that the target induces when it passes through a receptive field. When both variables remain constant across conditions, they should yield the same velocity threshold. This assumption is well supported by Experiment 5, which estimated a constant threshold of ~200 dva/s when endpoints were absent. Note that the expression of these thresholds in terms of absolute velocity, as requested by the reviewer, are already part of the manuscript (**Figure S5** in the **Supplementary Material**). In the main manuscript, we chose to report performance as a function of relative movement speed, which makes it much easier to compare performance across panels of the figure.

We have added the rationale of a fixed temporal-frequency threshold (that is independent of the movement distance) in response to this reviewer's point 2 in the introduction. Moreover, we now briefly point to this rationale when first introducing the hypotheses in the results section:

This paradigm allows for clear, mutually exclusive predictions: If visibility thresholds were simply a function of absolute movement speed (v) then they should be independent of movement amplitude (Figure 2e-f; *dashed white lines*). The rationale for this prediction is that, because of the fixed spatial frequency used in our experiments, speed is equivalent to temporal frequency, which predicts visibility thresholds for a wide range of visual stimuli^{21,45-48} (see **Introduction**). In contrast, if thresholds are a function of the kinematics of retinal motion during saccades, then they should systematically increase with movement amplitude, in proportion to the main-sequence. In that case, *relative* movement speed, expressed with respect to the expected peak velocity of a saccade ($v_{rel} = v/v_p$), should determine visibility (Figure 2e-f; *solid white lines*).

Minor comments:

1. The tables in the main text are not particularly useful and make the text heavier, I would suggest moving them to supplemental material.

We followed the reviewer's suggestion and moved the tables to the **Supplementary Material**.

2. Figure 4; it is not clear how the correlation values are calculated. Were they based on the gray points in the graphs or on the across subjects averages? If correlations are based on the gray points each subject is represented multiple times (it seems 5 eccentricities and 4 directions). This would make the points non-independent and would require some kind of adjustment when calculating the correlation.

We thank the reviewer for raising this question. First, we would like to clarify that the correlation coefficients in Figure 4 do indeed consider all gray data points. In fact, correlating subject averages would not be possible in this analysis, because it would not allow us to compare retinal and spatial directions—a key control analysis in our approach. Importantly, we account

for the fact that each observer contributed data points for each movement amplitude and direction in statistical analyses. Specifically, we performed stepwise regressions that show that the contribution of retinal speed is still significant when the impact of amplitude and direction are accounted for. We reported the results of this regression in the **Supplementary Material**, and highlight in the main manuscript that individual differences explain unique variance beyond that explained by movement amplitude and direction. Moreover, we now show that regression models using retinal direction as a predictor of visibility thresholds consistently outperformed models using spatial direction. This was true for both Experiments 3 and 4 consistent across analyses of speed and duration. Finally, we have added an analysis to section describing Figure 4 in the **Results** section (*Visibility covaries with saccade kinematics*), in which we remove the impact of amplitude on the correlation in Experiment 3. This analysis strongly supports our conclusion that visibility thresholds are correlated to the peak velocity of saccades in the opposite (i.e., retinal) direction.

The section for speed thresholds reads:

Note that significant correlations in Experiment 3 were not solely driven by the main sequence itself: Even after removing the main effect of saccade amplitude by removing the across-subjects mean of each of the dependent variables, we still found that saccadic peak velocity correlated significantly with visibility thresholds for absolute movement speed in the opposite (retinal) direction ($\rho = 0.40$, $p < 0.001$), whereas the correlations for the spatial direction vanished ($\rho = -0.07$, $p = 0.466$).

The corresponding section for duration thresholds reads:

Again, for the retinal direction in Experiment 3, these correlations remained significant even after removing the effect of saccade amplitude ($\rho = 0.41$, $p < 0.001$), while they did not for the spatial direction ($\rho = -0.05$, $p = 0.594$).

3. I find it a bit confusing that in the graphs relative speed is sometime defined as V/V_p , sometimes as V_p (which in the text is defined as peak velocity), and sometimes is referred as V_{rel} . Further, given that it is speed and not velocity, it would be better to call it S rather than V . Also, relative speed is not clearly defined in the text (see lines 82-86 vs. lines 129-131). Similarly, how was the absolute speed defined? It seems it is based on the peak speed for saccade of a given amplitude multiplied by a factor that is different for each amplitude. Why? This is also not clearly explained in the text.

We thank the reviewer for spotting these inconsistencies, which we have now fixed. To clarify, relative velocity v_{rel} is expressed in multiples of the peak velocity of saccades. We thus express absolute speed in units of v_p . In axis labels, units are provided in parentheses.

Moreover, we have further clarified that we used the same velocity factors for each movement amplitude. Figure 2d shows this for the example of a 4 dva movement. We have added this information directly to the figure and its caption.

Note that we have refrained from changing our labeling from v to S to stay consistent across parameters of saccades and stimulus movements.

4. In the graphs showing psychometric fits, are the dots with error bars average performance across subjects? Or do they represent performance when all subjects' trials are collapse together? If the former, reporting the number of trials in the graphs (e.g. 39745 trials) is not necessary. If the latter, it should be clearly specified in the text.

Error bars in these plots are 95% confidence intervals across subjects, which we have now clarified in the figure captions. We had reported the number of trials to give a sense of the considerable amount of data that went into each analysis. Following the reviewers suggestion, we have now removed that information from the figures.

5. It is not clear what Figure 2d shows, is it meant to show that with different stimulus durations relative speed is modulated in proportion to V_p , essentially resulting in longer stimuli duration be characterized by lower speeds?

Yes, the figure shows that when we varied relative speed for any given movement amplitude, it resulted in different movement durations. We have clarified this in the caption of that figure and added labels directly in that panel to clarify that each line corresponds to a certain relative movement speed.

Reviewer 3:

The manuscript describes that properties of perpetual invisibility of a moving stimulus can be predicted by kinematics of that moving stimulus on the retina. This is relevant for understanding saccadic omission, i.e., perceptual omission of stimulus during eye movements such as saccades. Through human psychophysics, the authors demonstrate that a shared law links the limits of perceiving stimuli moving at high speed to the sensory consequences of rapid eye movements. They develop a parsimonious model to describe how this effect takes place.

While the experiments are quite thorough, I do have some concerns regarding the interpretability of the results, which requires further clarification. I detail them below:

We appreciate the reviewer's positive summary and thoughtful comments. We have carefully addressed them below.

1. The authors use the term "lawfully" to describe the relation between peak velocity and duration of saccades with movement amplitude. Can the authors clarify what they mean by lawful in their manuscript?

We refer to the main-sequence relationship as lawful in the sense that it can be captured by a mathematical equation with a small number of parameters, such that different kinematic variables (amplitude, peak velocity, and duration) can be translated into each other in a predictable fashion. In that sense, the main sequence is a kinematic law much like Listing's law, Donders' law or the two-thirds power law in movement control, and it applies across all healthy human individuals (indeed, all known species that make saccadic movements, even including fruit flies; Fenk et al., 2022). In our work, we show that this lawful relation predicts the limits of visual sensitivity to stimulus movement. We have included this rationale when first introducing the main sequence in the manuscript:

Most prominently, the *main sequence* describes a lawful relation of saccadic speed and duration to the movement's amplitude: both peak velocity and duration of the movement increase systematically with the distance the eyes travel¹⁶ (Figure 1a). This relation is lawful in that the relevant kinematic variables (amplitude, peak velocity, and duration) mathematically relate to each other¹⁷⁻¹⁹, and it applies across all known species that make saccadic movements (even including fruit flies²⁰).

2. What is the basis of the null hypothesis used in this study, i.e. lines 122-125: "if visibility is simply predicted by absolute movement speed (v), then visibility thresholds should be independent of movement amplitude". But why should it be independent of movement amplitude? If saccadic omission relates to a sensory consequence, then one would have to think in terms of activity of visual neurons. Larger amplitude movement of a stimulus means that the stimulus will traverse several receptive fields. The duration each receptive field is exposed to should be larger than a critical number in order for that neuron to be active. For a constant speed, large amplitudes should therefore require longer total durations in order to be visible. This is consistent with the results shown in Fig. 3c which shows that the total duration increases with increasing amplitude. This (the solid white lines in Fig. 3c) would be the

expected behavior coming from a sensory consequence point of view. Maybe the authors can clarify the reasons/motivation for their null hypothesis.

We thank the reviewer for this important point. Accordingly, we have now clarified the rationale of the constant threshold predictions (see also our response to Point 9 of Reviewer 2) in two places, first in the Introduction:

A broad range of accounts has been put forward for this reliable absence of perceiving the sensory consequences of saccades^{10,35,36}, invoking mechanical^{37,38}, retinal^{24,39,40}, and extraretinal mechanisms^{31,41,42}. While there is consistent evidence for the saccade-locked reduction of visual sensitivity (especially to low spatial frequencies⁴²⁻⁴⁴ which should remain well resolvable at high saccadic speeds⁴⁵) perception of motion during saccades is well possible within a resolvable temporal-frequency range^{21,23,25}. These findings can be reconciled by a constant visibility threshold at some temporal frequency^{46,47} (or speed, for a given spatial frequency^{24,45,48}) beyond which a stimulus becomes invisible.

and then again when deriving predictions in the beginning of the Results section:

This paradigm allows for clear, mutually exclusive predictions: If visibility thresholds were simply a function of absolute movement speed (v) then they should be independent of movement amplitude (Figure 2e-f; *dashed white lines*). The rationale for this prediction is that, because of the fixed spatial frequency used in our experiments, speed is equivalent to temporal frequency, which predicts visibility thresholds for a wide range of visual stimuli^{21,45-48} (see **Introduction**). In contrast, if thresholds are a function of the kinematics of retinal motion during saccades, then they should systematically increase with movement amplitude, in proportion to the main-sequence. In that case, *relative* movement speed, expressed with respect to the expected peak velocity of a saccade ($v_{rel} = v/v_p$), should determine visibility (Figure 2e-f; *solid white lines*).

We also respectfully disagree with the reviewer's argument above. While it is true that larger movement amplitudes will result in the involvement of a larger number of receptive fields and that each receptive field will require a certain input intensity (as in contrast or duration) to respond, it does not necessarily follow that large amplitudes should require longer presentations to be visible. That is, because the mere response of one single receptive field along the target's trajectory could in principle be sufficient to signal its presence in the upper or lower visual field. If visibility were only determined by the duration of stimulation that a single receptive field is exposed to, then this duration is the same across amplitudes as long as the target's velocity remains the same. In this case, all receptive fields that the stimulus passes through should have the same probability to respond. This case occurs in Experiment 5, which is shown in Figure S5 in the **Supplementary Material**: In the absence of end points it is indeed absolute velocity that determines task performance, irrespective of movement amplitude. The idea of a constant temporal-frequency threshold is well supported by studies of human temporal sensitivity (see above) and constitutes a parsimonious null hypothesis for our investigation.

3. If visibility thresholds depend on duration a stimulus is exposed, then perhaps looking at how visibility thresholds vary with the duration of exposure per unit amplitude might yield some

interesting insights. For example, does the duration of exposure scale linearly with total amplitude or not?

This is an interesting question, and we believe that the answer is already included in our analyses. The threshold duration varies linearly with amplitude (e.g., Figure 3c), such that the exposure per unit amplitude corresponds to the slope of this linear increase. Our data suggest, therefore, that this is a constant, which may however vary across endpoint durations. Specifically, the slope of this largely linear increase of threshold durations is at 5.7 ms/dva for 0 ms, decreases to 3.75 ms/dva for 12.5 ms, and settles at 3.2 ms/dva for 50 ms and beyond. Note that the latter estimate is very close to the slope used to model the main-sequence relationship in the stimulus (i.e., 2.7 ms/dva; see Methods section, Procedure in perceptual trials), and is proof that—provided endpoints are available—duration thresholds indeed follow the relationship described by the main sequence.

4. Fig. 5a shows that in case there are no end-points, then visibility thresholds do not depend on the saccade main-sequence. So it appears that end-points are critical not just for rendering the motion invisible, but also in a way that it aligns with the main-sequence. But the way the paper mostly reads is that kinematics of retinal motion shape the visibility thresholds. Can the authors disambiguate this?

We thank the reviewer for this critical remark. Indeed, our results show that (even very briefly presented) endpoints were necessary to establish the main-sequence relationship described in this paper. Yet, their exact role is unclear. On the one hand, endpoints could serve as metacontrast masks (Macknick & Livingstone, 1998), capable of masking retinal stimulus traces across considerable spatial distances (cf. Duyck, Collins, & Wexler, 2016). While this mechanism is without doubt operational in our case, it alone does not explain the main-sequence dependence we observe: A mask that is constant across experimental conditions would be expected to be equally detrimental across all of our tested conditions. On the other hand, endpoints serve as a visual reference, based on which the system can estimate movement amplitude. It is known from a range of motion-related phenomena and illusions that visual localization in the face of motion is highly prone to error (for an overview, see Whitney, 2002). This might be even more true for motion at saccadic velocities, close to the detection threshold. In the absence of static endpoints, even though it was still possible to ascertain the presence of a moving target, it was indeed hard to determine motion amplitude in these trials, especially as amplitudes were displayed in a randomly interleaved manner. In short, while we cannot explicitly disambiguate the two effects, our present work operates under the assumption that the effect is twofold: While static endpoints—a ubiquitous part of the saccade-induced translation of objects across the visual field—may serve as masks for the residual motion signal, they enable an evaluation of retinal kinematics in terms of the main sequence. We have included this consideration briefly in the final paragraph of the section reporting Experiment 5 (*Main-sequence relation requires static endpoints*):

Here, the effect of endpoints appears to be twofold. On the one hand, endpoints may have served as masks⁶⁵, capable of eliminating retinal stimulus traces even across considerable spatial distances³³. On the other hand, because localization of moving stimuli is error-prone⁶⁶, the

endpoints may have served as visual references, improving access to the movement amplitude, which is essential for evaluating retinal trajectories in terms of the main sequence.

5. In the current study, constant stimulus velocity is used, meaning there is infinite acceleration/deceleration from the endpoints. It is likely that the acceleration/deceleration profile of end-points modulate the visibility thresholds given that they play a crucial role. Previous studies (eg Macknik and Livingstone 1998) have highlighted that the duration between stimulus and endpoint is relevant in rendering the stimulus invisible. Do the authors expect the velocity profile to modulate visual thresholds? Does this change any of their conclusions?

We were also curious what acceleration/deceleration of saccadic velocity profiles would affect visibility (as were Reviewers 1 and 2) and had specific predictions, which we tested in a new, pre-registered experiment. We added this new experiment (Experiment 2b) to the **Supplementary material**, and describe its main results in the main text. We provide a brief summary and the new **Supplementary Figure S3** in response to Point 1 of Reviewer 1.

We also thank the reviewer for pointing us to this relevant reference, which we now included in the manuscript.

6. The finding that visibility covaries with saccade kinematics is very intriguing. The results here are presented as a support for the claim that visibility thresholds strongly depend on kinematics of retinal motion. Can the authors comment on why they think this is a stronger claim than the notion that saccade kinematics are tuned to render the motion invisible.

We thank the reviewer for raising that point. We do not think this is necessarily a stronger claim, but it was our starting point. We designed (and pre-registered) all experiments and hypotheses from the perspective that the visual system exploits the lawful relationship between saccade amplitude and its retinal consequences for high-speed perception. This idea is consistent with the finding that oculomotor control of saccades is well-developed in babies at 2 months of age or earlier (Garbutt et al., 2007). It is also consistent with recent findings that altering the visual consequences of saccades quickly results in adjustments of trans-saccadic perception (Zimmermann, 2020). Nevertheless, we expanded the paragraph that discusses that our results could be driven by different underlying causal directions.

From the outset, we hypothesized that visibility thresholds are the result of a lifetime of exposure to saccade-induced retinal motion¹⁰, which follows the main sequence in babies after two months of age or earlier⁷⁷. Indeed, the human visual system may never experience motion over finite distances at speeds higher than those imposed by saccades, such that its sensitivity is limited to that range. This direction of causality (i.e., that movement kinematics impact visual sensitivity) is consistent with recent findings that changes in the visual consequences of saccades results in quick adjustments of visual sensitivity during saccades³¹. A complementary view of our data (that would be equally adaptive) is that saccadic speeds are tuned to exploit the properties of the visual system, much like saccade amplitudes appear to be adapted to receptive field sizes and adaptive properties of neural populations in a range of different species⁷⁸. Indeed, the kinematics of eye movements are reliable over time and across experimental conditions^{56,59}, and there are a

number of striking examples showing that the oculomotor system resorts to keeping the kinematics of its movements relatively constant. For instance, patient HC, who could not move her eyes from birth, readily moves her head in saccade-like movements⁷⁹. Similarly, humans whose head is slowed down by weights put on the head compensate for these external forces to regain the velocity-amplitude relation of their combined eye-head movements⁸⁰. Finally, gaze shifts of a certain amplitude have similar dynamics, even if the eye and head movements that contribute to them have a very different composition⁸¹. These data suggest that the saccadic system aims to keep the kinematics of saccades constant over a large range of conditions. In the light of the data presented here, a possible function of this would be to keep movement kinematics in a range that yields perceptual omission of the saccades' retinal consequences, while maintaining a high degree of sensitivity to high-speed motion. Understanding how such an alternative causal direction (or an even more complex causal structure) might underlie our observed results is a key question that follows from the lawful relation between action and perception that the present study revealed.

7. The model replicates the experimental findings. However in its current state, I find the model not very informative. For example, can the authors elaborate on where/or what processes in the brain might the different model components be analogous to? And how can the model be used to generate testable hypotheses for future studies? What do we actually learn from this model?

The reviewer is right about the fact that the model does not assume specific processes or regions in the brain. Up to this point, we have deliberately omitted any discussion of the responsible mechanism or its neural origin, as most of it would have been speculation. The additional simulations conducted to determine under what circumstances the described model behavior arises (see 'Grid-search analyses' in the **Supplementary Material** of the revised paper), however, suggest clearly that the most crucial parameter of the model is the scale of the Gamma-shaped temporal response function θ_{RF} , which describes the amount of visual persistence. Such intensity-dependent persistence occurs at every stage of visual hierarchy, even as early as in the retina. Initial model parameters were informed by Schweitzer et al. (2023), who fitted post-stimulus response histograms recorded from simple cells in primary visual cortex with the Gamma function and found that a scale parameter of around 20 closely approximated temporal response patterns across a range of contrasts. The finding that this scale value is extremely close to those model parameters that produce the best fits to our experimental results (see **Figure S6b-c**) suggests that experimental results could well have arisen from the temporal dynamics in primary visual cortex, or earlier. An additional relationship can be found for the SD of the Gaussian spatial kernel σ_{RF} . That is, Anderson and Burr (1987) estimated the Gaussian SD of spatial receptive fields with preference for 1 cycle per degree to amount to 0.3 cpd. In our simulations, best fitting σ_{RF} were also found below 0.5 cpd, suggesting that this model characteristic, too, could have been grounded in early visual coding of orientation and motion. We have included these considerations in the section *Grid-search analyses* in the **Supplementary Material**, and refer to it in the revised manuscript.

With these parameters in mind, the model could well make predictions about the outcome of other experimental manipulations. For one, both receptive field sizes and temporal response functions are known to vary across spatial frequencies. For instance, using a Gabor patch with

lower spatial frequency would result in less sluggish temporal response functions (e.g., Frazor, Albrecht, Geisler, & Crane, 2004). In terms of the model, this would lead to higher evidence for motion and, provided that the noise-level of the readout remains the same, result in higher velocity thresholds – a prediction that we have already achieved preliminary experimental evidence for. An intriguing outlook would be to relate temporal response functions in different species to their perceptual abilities.

Finally, what we can learn from this model is, in principle, our experimental finding could—most parsimoniously—be understood as arising from a combination of the specific motion trajectory and the temporal dynamics of the early visual system. Given that the latter are well known, the key to explaining the intriguing pattern of results at hand must lie in the motion trajectory itself, and in the main-sequence relationship. With suitable extensions, such as orientation-selective channels and temporal response functions that reflect temporal sensitivity profiles at different spatial frequencies, the model may provide a valuable tool to predict and understand results from future experiments using this or similar tasks.

8. In the introduction and discussion, the authors mention that visibility thresholds relate to kinematics of retinal motion and this could be a consequence of eye movements. Do the authors mean that this relation is learnt over a human's lifetime or do they mean that this relation has evolved in species with eye movements? This could be a potential pointer for future studies to explore. For example, if it is learnt over a human's lifetime then presumably age of participants should matter. On the other hand if it evolved in species with eye movements then it would be interesting to see how visual thresholds relate to kinematics in species with less frequent eye movements

These are fascinating questions and, indeed, we would predict such correlations. We included a few examples in the final paragraph of the revised Discussion:

We propose that such coupling is not limited to the human visual system, but should apply across species (e.g., faster-moving animals should be more sensitive to high-speed motion) and sensory modalities (e.g., auditory motion perception may be constrained by the kinematics of head movements}, provided that the actions sampling the environment impose regularities onto the input of the sensory system.

Reviewer 1:

The authors should be commended for a rigorous and thoughtful response to previous comments. They have addressed my concerns and I am happy to recommend publication.

We thank the reviewer for their very positive feedback.

Reviewer 2:

The authors did an excellent job in addressing the reviewers' concerns. The abstract has been revised and it is now more informative and links to previous work, before missing, have been highlighted. Most importantly, the authors conducted another experiment in which stimulus motion mimicked saccade-induced motion confirming the main conclusion.

We thank the reviewer for their great appreciation of our work on the revision.

I only have a couple of comments:

1. Given their importance in ruling out possible confoundings, the relatively large sample size, and the fact that Exp 2b replicates conditions of retinal motion that more closely resemble what experienced during actual saccades, and the fact that they directly refer to saccadic omissions, I think the supplementary results of Figure S3 (Exp 2b) should be included in the main text, and it should be explained why the results for relative movement speed do not perfectly match with the predictions. Ideally, Exp 2b should be the main experiment. It would also be important to perform analyses similar to those of Fig 4 for experiment 2b.

We agree that the new Experiment 2b is important for our conclusions, and we understand the inclination to include it in the main text. At the same time, we feel that this would only be possible in tandem with Experiment 2a without losing coherence. This is because Experiment 2b differs not only in the velocity profile of the stimulus' movement but also in the task observers were given (detection rather than curvature discrimination) from all other experiments in the main text. We fear, however, that completely moving both experiments to the main text would make the manuscript even denser, at the expense of readability. We thus opted for a middle ground by extending the description of the results of Experiments 2a and 2b in a dedicated subsection in the main text (*Replication in detection tasks and for time-varying velocity*) and adding a summary figure that combines results from both experiments (new Figure 4). A more comprehensive description of the results of Experiments 2a and 2b (including dedicated figures) remains in the Supplementary Material.

Note that an analysis analogous to Figure 4 (now Figure 5) in the manuscript is not possible for the new data set, as we did not collect eye movement data for the participants of Experiment 2b. Generally, given the close association between visibility thresholds and movement amplitude, however, a correlation between saccade kinematics and visibility threshold would again be expected (presumably with a steeper slope than those in what is now Figure 5a, because of the steeper slope in visibility thresholds for absolute velocity).

2. The authors mention that the retinal motion introduced by post saccadic lens wobbling is negligible, however Taberero and Artal analyses, estimating the actual retinal motion on the cone mosaic resulting from lens wobbling, suggest otherwise (max retinal displacement of 0.3 deg for a 9 deg saccade). This wobbling is particularly consequential for the stimulus the saccade lands on, the stimulus at the very center of gaze. Here, cone receptors are most closely spaced, and even a small amount of motion can have important perceptual consequences. I am not asking the authors to replicate this motion with an additional control experiment, but they should at least acknowledge and discuss the potential relevance of this retinal motion component that is completely ignored in their model and in their simulated motion trajectories. How can this additional component of retinal motion influence perception and possibly the pattern of results shown here?

We thank the reviewer for bringing up this important issue again, on which we are happy to elaborate. We are indeed aware of the findings of Taberero and Artal (2014), which match well with the estimate made by Deubel & Bridgeman (1995a), that is, a retinal shift of 0.4 dva for 12 dva saccades. With an assumption of linearity, both studies would thus estimate a retinal shift of 0.033 dva per degree of saccade amplitude. Yet, psychophysical results suggest that—even in foveal vision—this retinal shift need not fully translate to perceptual consequences (Deubel & Bridgeman, 1995b). To directly relate to the reviewer’s concern, we would like to point to a recent preprint of ours (Figure 8d in Schweitzer, Doering, Seel, Raisch, & Rolfs, 2023; <https://doi.org/10.1101/2023.03.15.532538>) where we left post-saccadic oscillations (PSOs) in simulated saccade trajectories intact and compared their impact on post-saccadic task performance during simulated and real saccades. Our analyses revealed that larger PSO amplitudes indeed led to a decrease in task performance, but much more drastically in simulated than in real saccades, suggesting that the perceptual effect of PSOs for saccades is smaller than what would be expected from PSO amplitudes alone. To estimate the perceptual consequences of PSOs measured by video-based eye tracking, we proposed a mathematical approximation that – when applying its predictions to our simulated-saccade data – provided a remarkable fit to the task performance measured with real saccades. Thus, when PSOs are used in simulated saccade experiments, their effective retinal consequences must be considered. We currently lack a computational model that approximates these retinal consequences based on saccadic trajectories. Fortunately, while these consequences can be relatively large in post-saccadic foveal vision, they would only have affected the (peripheral) end position of the stimulus in our experiments. To acknowledge these points, we have extended the corresponding section in the manuscript as follows:

Note that the simulated movements did not include post-saccadic oscillations, which are able to introduce retinal image shifts of approximately 0.033 dva per degree of saccade amplitude^{93,94} (due to inertial motion of the crystalline lens), but also entail measurable perceptual consequences^{34,95}. Compared to real saccades, our simulated saccade profiles—assuming our largest movement amplitude (i.e., 12 dva) and a corresponding lens overshoot of 4 dva⁹³—may thus have underestimated the target stimulus’ post-saccadic movement amplitude by up to 0.4 dva. Due to the difficulty involved in modeling the nonlinear dynamics of post-saccadic oscillations and, more critically, their retinal

consequences^{61,62}, and because our stimulus endpoints were in peripheral rather than foveal locations, we chose not to include this aspect in the stimulus trajectories of Experiment 2b.

Reviewer 3:

I thank the authors for their detailed replies and explanations. The manuscript definitely has improved.

We thank the reviewer for appreciating our previous revision. We address the remaining comment below.

In reference to my comment #4 on the previous submission, that end-points appear to be critical not just for rendering the motion invisible, but also in a way that it aligns with the main-sequence. I understand that in the current experimental setting, it is not possible to explicitly disambiguate the extent to which end-points play a role in rendering motion invisible in a way dependent on the main sequence. The authors have now addressed this briefly in the relevant results section. It will be helpful if the authors make this limitation and their underlying assumptions and its justification more explicit in the discussion section.

We have taken the opportunity to acknowledge this point to the Discussion section of the manuscript, where we now write:

Even though our experiments up to this point were not capable of explicitly disambiguating the possible roles of endpoints as either pre- and post-movement masks or as visual cues for movement amplitude, the tight correlation of this conjunction to the main sequence relation of saccades is striking.